# Strategic Planning and Rationalizing on Trees Make LLMs Better Debaters

**Danqing Wang**[†]**, Zhuorui Ye**[†] [∗]**Xinran Zhao**[†]**, Fei Fang, Lei Li**
Carnegie Mellon University
{danqingw,xinranz3,feifang,leili}@cs.cmu.edu
cuizhuyefei@gmail.com

## Abstract

Winning competitive debates requires sophisticated reasoning and argument skills. There are unique challenges in the competitive debate: (1) The time constraints force debaters to make strategic choices about which points to pursue rather than covering all possible arguments; (2) The persuasiveness of the debate relies on the back-and-forth interaction between arguments, which a single final game status cannot evaluate. To address these challenges, we propose `TreeDebater`, a novel debate framework that excels in competitive debate. We introduce two tree structures: the **Rehearsal Tree** and **Debate Flow Tree**. The Rehearsal Tree anticipates the attack and defenses to evaluate the strength of the claim, while the Debate Flow Tree tracks the debate status to identify the active actions. `TreeDebater` allocates its time budget among candidate actions and uses the speech time controller and feedback from the simulated audience to revise its statement. The human evaluation on both the stage-level and the debate-level comparison shows that our `TreeDebater` outperforms the state-of-the-art multi-agent debate system, with a +15.6% improvement in stage-level persuasiveness with DeepSeek and +10% debate-level opinion shift win. Further investigation shows that `TreeDebater` shows better strategies in limiting time to important debate actions, aligning with the strategies of human debate experts. [1]

## 1 Introduction

Competitive debate is a structured battleground of arguments. It plays a crucial role in domains such as education (Huryn, 1986; Ball, 2021), legislation (O'Connell, 2002), and politics (Hart & Jarvis, 1997; Holbrook, 1999). In a debate competition, two sides argue on the same motion under strict time limits. It requires complex reasoning skills and precise selection of impactful arguments. Expert debaters allocate time wisely among arguments in different stages. Unlike other tasks, there is no ground-truth answer in a debate: The winner is determined by who sways the audience or judges more effectively. Research shows that success in debates depends heavily on a debater's ability of intense back-and-forth competition and on-the-fly decision-making: predict, respond to, and adjust based on their opponent's arguments (Rapoport, 1960; William L. Benoit & Verser, 2003; Salvi et al., 2024). Although large language models (LLMs) are effective in generating arguments, human evaluators continue to rate AI debaters as less convincing than human opponents in live debates (Flamino et al., 2025). These gaps highlight that current LLMs struggle with planning effective arguments in the dynamic back-and-forth interaction of competitive debate.

The main challenges of debate AI lie in strict time limits and the lack of objective reward signals. First, the timed setting makes it impossible for LLMs to generate verbose arguments for each candidate action. For example, there can be a trade-off between attacking the opponent's claims and defending one's own claims. A formal competitive debate typically consists of three stages: an opening statement, a rebuttal statement, and a closing statement. In each stage, the debater is required to carefully allocate their limited speaking time to propose new claims, reinforce the existing claims, attack the opponent's claims, or rebut the opponent's attack. The debaters must prioritize

---

[∗]Part of the work was done when Zhuorui was a visiting intern at Carnegie Mellon University.

[1]Code and data have been released at https://github.com/LeiLiLab/TreeDebater.

the most impactful actions and dynamically adjust their plan as the debate progresses. However, unlike games such as Go (Silver et al., 2016; 2017) or Werewolf (Xu et al., 2023; 2024), there are no rule-based winning conditions. Instead, the winner depends on the evolving flow of arguments and counterarguments between the two sides. This dynamic nature makes it difficult to plan based on the final reward signals.

In this work, we propose to model the dynamic interaction on trees and show that planning on these trees can significantly improve the debating capacity of LLMs. Human debate experts implicitly employ tree-like reasoning: they rehearse potential rebuttals before the debate starts by anticipating the claims an opponent might say and formulating the potential responses for each of them in a tree format. During the debate, they track the flow of the debate with another tree to keep a structured mental map of which points have been addressed or remain standing. Inspired by these human strategies, we introduce a new debate system `TreeDebater` to help LLMs make tactical decisions in the debate. Akin to human debaters, we introduce two kinds of trees in `TreeDebater`: **Rehearsal Tree** to hypothesize the opponent's attacks in advance, and **Debate Flow Tree** to track the debate status. Specifically, in the Rehearsal Trees, `TreeDebater` scores arguments based on the potential tree-structured attacks and defenses. During the debate, `TreeDebater` keeps track of the debate status through the Debate Flow Trees and provides the candidate action set. `TreeDebater` retrieves prepared arguments from the Rehearsal Tree for each candidate action to draft the statement. Finally, `TreeDebater` revises its statement based on simulated audience feedback and uses a speech-based search algorithm to fit the statement into the speaking time limitation.

To evaluate the effectiveness of `TreeDebater` in competitive debate, we carried out extensive experiments with the multi-agent debate system Agent4Debate (Zhang et al., 2024), which has shown comparable results to human debaters. We design two kinds of human evaluation to compare the debate performance: **stage-level head-to-head comparison** with a fixed debate context, and **debate-level end-to-end comparison** with an Oxford-style debate format. Human evaluation demonstrates that `TreeDebater` consistently outperforms the baseline in the dimension of the average persuasiveness score and the stage-level and debate-level win rate. It is preferred 1.5x and 3.5x over the baseline in the stage-level and debate-level comparison with the Gemini-2.0-flash backbone, where the gain also generalizes to another backbone, DeepSeek-V3.

To conclude, our contributions are listed as follows:

- We develop a sophisticated AI system for interactive and time-constrained competitive debate. It includes two tree structures: a Rehearsal Tree and a Debate Flow Tree to prepare arguments and plan actions in debate.
- We incorporate a speech time controller to properly control the speech time and a simulated audience into the `TreeDebater` to refine the debate draft into a human-like statement.
- We conduct extensive human evaluation on the stage-level and debate-level. Our study shows that `TreeDebater` is more persuasive than the prior multi-agent debate system. Further investigation shows that `TreeDebater` elicits more diverse actions during the debate with a better logic flow and emotion tone, similar to human debate experts.

## 2 RELATED WORK

**Computational Argumentation.** Previous studies used computational methods to study argumentation processes, such as argument mining (Lawrence & Reed, 2019), the polarity of arguments (Agarwal et al., 2022; Liu et al., 2021; Zhao et al., 2021), argument structure (Zhang et al., 2016; Li et al., 2020), and argument generation (Hua et al., 2019; Lin et al., 2024). Slonim et al. (2021) built the first autonomous debating system, Project Debater, with four argumentation modules: argument mining, argument knowledge base, argument rebuttal, and debate construction. While the first three modules have shown superior performance in creating high-quality arguments, the debate construction is based on a human-defined template, which cannot adapt to the dynamics in competitive debate. Instead of generating a better argument, our work focuses on the decision-making in the debate, such as which point to argue.

**Debate in Large Language Models.** There are two research directions in debate with large language models. One is to enhance LLMs' capability in solving problems by debating with different solutions, such as debate for reasoning (Du et al., 2023; Liang et al., 2023; Yin et al., 2023), evaluation (Chern

et al., 2024; Chan et al., 2024), or safety (Irving et al., 2018). Unlike competitive debating, these problems have an optimal solution, and the purpose of the debate is to find this solution by discussion. Another line of work uses LLMs' argumentation capability to enhance the debating performance. For example, Lee et al. (2023) uses role-playing to simulate debate between different populations. Zhang et al. (2024) design a multi-agent framework to work collaboratively during the debate, showing comparable performance with human debaters. Different from previous work, we consider the strict time limitation in competitive debate, which requires LLMs to make tackle decisions and allocate the time budget to the most important actions.

**Language Agents for Games.** Besides debating, recent studies also investigate language agents in other strategic games that require strong communication between players, such as Diplomacy (Meta et al., 2022), Werewolf (Xu et al., 2023; 2024), and Avalon (Wang et al., 2023). Cicero (Meta et al., 2022) and Xu et al. (2024) learn a policy to choose suitable strategies during the game, while Xu et al. (2023) and Wang et al. (2023) design deliberate prompts to improve the performance. These games have a clear definition of winning. For example, the winning condition of villagers in Werewolf is to eliminate all the werewolves, while the winning condition of werewolves is to kill all villagers. These clear winning decisions provide an objective reward signal for learning the strategies. However, there is no clear winner in competitive debate, making the strategy learning more difficult.

# 3 STRATEGIC PLANNING IN COMPETITIVE DEBATE

We first introduce the background of competitive debate (Section 3.1). Then we propose our debate system `TreeDebater`, which leverages Rehearsal Tree (Section 3.2) to anticipate the back-and-forth between sides and prepare arguments before the debate starts. During the debate, it tracks the flow with the Debate Flow Trees (Section 3.3) to strategically plan among the candidate actions. Furthermore, `TreeDebater` incorporates human debate trees (Section 3.4) and a speech time controller (Section 3.5) to refine statements based on human-aware feedback and time constraints. An overview of `TreeDebater` is shown in Figure 1.

## 3.1 COMPETITIVE DEBATE

Following previous work on the debate game, we focus on the simplified Oxford-style debate (Slonim et al., 2021; Zhang et al., 2016; 2024). It is a competitive debate format where two sides argue for (**Pro** stance) and against (**Con** stance) a specific motion. There are three stages. In **Opening Stage**, each debater has 4 minutes to deliver their main claims and arguments to support their assigned stance. In **Rebuttal Stage**, the debaters take turns attacking the opponent and defending themselves in 4 minutes. In **Closing Stage**, the debaters summarize the debate in 2 minutes.

Audience members are asked to vote before and after listening to the debate. The final winner of the debate is the one with more votes swaying towards it. We call this win as **Opinion Shift Win**. There are several other debate terms. For example, a *claim* $c$ is a proposition about the motion, the *argument* $x$ uses reasoning or evidence to support the claim, and *statement* is the whole passage presented in one debate stage. There are four typical *actions* that the debater can take in each stage: *propose* a new claim, *reinforce* the existing claims, *attack* the opponent's claims, and *rebut* the opponent's attack. A *battlefield* refers to a conflict zone in the debate, where two sides contest on a specific point. It includes several rounds of proposal, attack, and defense.

## 3.2 REHEARSAL TREE

Before the debate begins, human debaters usually prepare for the potential attacks and defenses of their claims. This rehearsal can help them retrieve relevant evidence and evaluate how robust their claims are towards the attack. Motivated by that, we propose the Rehearsal Trees $T_r$ to anticipate the back-and-forth between two sides.

**Build Rehearsal Tree via Top-Down** Specifically, `TreeDebater` proposes $n$ candidate main claims $C = \{c_0, \cdots, c_n\}$ and constructs a Rehearsal Tree $T_r$ with the maximum depth $L$ for each claim $c = x^{(0)}$. Each node is an argument $x$, and its children are the potential counterarguments. Thus, each node is on the same side of its grandparent. We define the attack score of the $l$-th level node $x^l$ as its attack impact towards its parent $x^{l-1}$, which is $r_a(x^l, x^{l-1})$. The support score of $x^l$

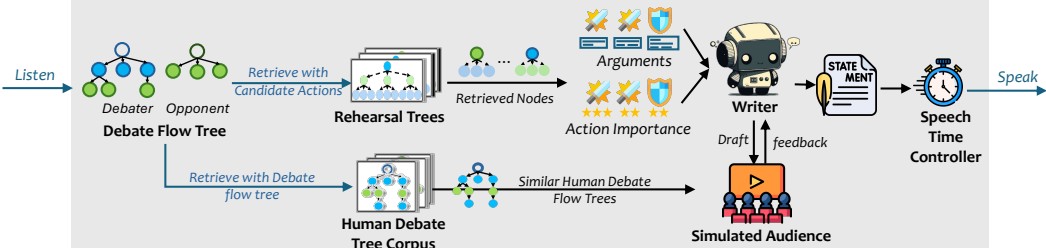

Figure 1: The overall workflow. In each stage, `TreeDebater` (i) updates two Debate Flow Trees with extracted action tuples from the current statement; (ii) retrieves prepared arguments from the Rehearsal Tree for the candidate actions; (iii) generates a draft based on the retrieved arguments and important scores; (vi) lets simulated audience provide feedback based on retrieved human debate flow tree; (v) revises based on feedback from the simulated audience and speech time controller.

is its support impact towards its grandparent $x^{l-2}$, which is $r_s(x^l, x^{l-2})$. Here, $r_s$ and $r_a$ are two scoring models that evaluate the impact between two arguments. For the main claim $x^0$ of the root, its support score is defined as its support in the assigned stance $s$.

**Calculate Strength Score via Bottom-Up** To evaluate the utility of an argument in the debate for our side comprehensively, we define a strength score $f_k(x^l)$, which considers both the support and attack score of the node $x^l$ and the influence of its descendants within depth $k$ ($k$-subtree). For $k = 0$, the strength score can be represented as:

$$f_0(x^l) = \begin{cases} r_s(x^l, s) & \text{if } l = 0 \\ r_a(x^l, x^{l-1}) & \text{if } l = 1 \\ \frac{1}{2}(r_a(x^l, x^{l-1}) + r_s(x^l, x^{l-2})) & \text{if } l \geq 2 \end{cases} \tag{1}$$

For $k > 0$, we define the strength score recursively, adopting a minimax-like perspective from our side's point of view. It represents the minimum utility our side can expect from argument $x^l$, since we assume that the opponent will always choose the counterargument $x^{l+1} \in \text{Child}(x^l)$ that maximizes their utility, which corresponds to minimizing our utility in this zero-sum debate. Therefore, the a $k$-step strength score can be represented as:

$$f_k(x^l) = f_0(x^l) - \gamma \cdot \max_{x^{l+1} \in \text{Child}(x^l)} f_{k-1}(x^{l+1}), \tag{2}$$

where $\gamma$ is the decay coefficient (which is 0.8) since one may choose to stop further reacting, and $\text{Child}(x^l)$ is the set of $x^l$'s child nodes. The whole process of constructing the Rehearsal Tree is illustrated in Alg. 1.

To conclude, the Rehearsal Tree anticipates the attack and defense for each main claim in a tree format and calculates the $k$-step strength score to evaluate the utility of the claim. We further let `TreeDebater` propose candidate claims and construct the Rehearsal Trees for the opposite side $T_{r_{oppo}}$, making it possible to prepare for the attack and defense of the opposite side.

### 3.3 DEBATE FLOW TREE

The dynamic interaction in debates makes it easy to lose track of the back-and-forth of points. We propose the Debate Flow Tree $T_d$ to record the debate status, simulating the note-taking of humans.

**Maintain Debate Flow Tree** The Debate Flow Tree tracks the debate status by keeping all proposed claims with the corresponding attack and defense in a tree structure. The tree node consists of the claim, the arguments to support the claim, the state (*proposed* or *attacked*), and the number of visits. Each time after listening to one statement, `TreeDebater` first extracts the tuples of (action, claim, argument, target) from the statement, and then updates the Debate Flow Tree with these tuples. If a new claim occurs, a new node with *proposed* state will be created under the tree root.. If an existing claim is attacked, its target claim node will change to *attacked* state, and a child node will be created to indicate this attack. When new arguments are created to reinforce an existing claim, the corresponding claim node will be updated with these new arguments. The update process is described in Alg. 2 and Figure 4b.

**Extract Candidate Actions from Debate Flow Tree** `TreeDebater` can filter out the candidate actions it can take in the current stage of the debate based on the Debate Flow Tree. The criteria

---

**Algorithm 1:** Build Rehearsal Tree

---

**Input:** Root claim $x^0$, Attack Scorer $r_a$, Support Scorer $r_s$, Maximum branch $B$, Maximum depth $L$

**Output:** Rehearsal tree $T_r$

---

1   Initialize the Rehearsal Tree $T_r$ with the claim $x^0$;
2   Initialize queue $Q$ with $T_r$'s root note;
   // Phase 1: Build Rehearsal Tree via Top-Down
3   **while** *Q is not empty* **do**
4      Extract the first node $x^l$ from queue $Q$ with the level $l$ ;
5      **if** $l < L$ **then**
6         Generate $B$ children using LLM and retrieve evidence for them ;
7         Calculate $r_s$ and $r_a$ for each child ;
8         Add children to $Q$;
   // Phase 2: Calculate Strength Score via Bottom-Up
9   **for** *level* $l \leftarrow L$ **to** *1* **do**
10      **foreach** *node* $x^l$ *at level l* **do**
11         **for** $k \leftarrow 0$ **to** $L - l$ **do**
12            Calculate $f_k(x^l)$ using Eqn 1 and 2;
13   **return** $T_r$;

---

include: (a) propose only at the opening stage; (b) rebut the latest nodes (leaf node) of the opposite side; (c) reinforce the nodes of its own side; (d) attack the nodes of the opposite side.

**Retrieve Prepared Arguments from Rehearsal Tree** `TreeDebater` retrieves the arguments prepared for this action from the Rehearsal Trees after getting the candidate actions. For example, to propose or reinforce a claim, we should retrieve the arguments that support this claim or oppose its counterclaim. To attack or rebut a claim, we should retrieve its counterclaim. `TreeDebater` also retrieves the $k$-step strength score of the claim, and here $k$ depends on the number of rounds remaining in the debate. For example, there are 3 remaining rounds (Con's opening, Pro's Rebuttal, Con's Rebuttal) [2] for the Pro side in the opening stage, which means that there can be at most 3 levels of child nodes. Therefore, we use $k = 3$ to get the strength score. We use one of the state-of-the-art embedding models, Gemini-text-embedding-4 to get the embedding of the claims. If the cosine similarity of two embeddings is higher than a threshold of 0.8, we view these two claims are similar. To find the most similar tree node, we traverse all tree nodes and calculate the cosine similarity between the claim of the node and the target claim. Details of the retrieval process are given in Appendix D.

To conclude, the Debate Flow Tree tracks the debate status and provides the candidate actions. These candidate actions are enhanced with prepared arguments and the $k$-step strength score from the Rehearsal Tree. These help the LLM writer select the impactful actions with strong arguments and evidence to draft the statement.

### 3.4   AUDIENCE FEEDBACK BASED ON HUMAN DEBATE FLOW TREE

We provide simulated general audience feedback by evaluating different key aspects of the speech based on the retrieved human Debate Flow Trees. We first curate a human debate flow tree corpus with Debate Flow Tree created from human debate corpus. During the debate, we reformulate the current Debate Flow Tree into a tree-like string. Then we employ semantic search based on the embeddings of the debate flow tree string. Similarly to the argument retrieval, Gemini-text-embedding-4 is used to get the embedding of the tree-like string of the current debate flow tree and of trees in human debate flow tree corpus. The top 1 human debate flow tree is retrieved with a similar threshold of 0.8. We add the retrieval to the instruction for simulated audience to give the audience a better sense of the back-and-forth and the statement styles in the human debate competition.

---

[2] Since the Closing Stage only summarize what have been said, we do not view it as an effective round which can take debate action.

The simulated audience provide concrete and focused feedback on important aspects in debate, such as *message clarity*, *engagement impact*, *evidence presentation*, and *persuasive elements*. By revising based on the audience feedback, TreeDebater can learn the allocation pattern and persuasive elements embedded within relevant human debate structures. Details of the construction and retrieval of human Debate Flow Trees are in Appendix D, and an example of the resulting audience feedback is in Table 15.

## 3.5 SPEECH TIME CONTROLLER

With the selected actions from the Debate Flow Tree, our hope is then to generate a coherent speech within a reasonable time range $[t_l, t_r]$. When drafting the statement, we incorporate the word budget into the instruction, which is based on a rough estimate of the speaking time, i.e., 130 words per minute. However, we find that it is difficult for LLMs to follow the length constraint (Jie et al., 2024; Butcher et al., 2025), and the number of words cannot control the precise speech time since each word takes a different speaking time due to its phoneme and might be affected by emotion. To ensure that the speaking time of the statement fits the precise time limitation in the competitive debate, we introduce a speech time controller in TreeDebater to provide a better estimation of the speech length and guide the LLM writer in revising the statement for the time restriction.

We use a light-weight text-to-speech model, FastSpeech (Ren et al., 2020), to estimate the speech length. At each iteration, the speech time controller converts the current statement to audio and calculates its time cost. The calculated time cost $t$ and the word budget $n$ in the instruction of this statement will be used to search for a new word budget for the revision. The iteration will stop when the speech time falls into the suitable range or it reaches the maximum revision limit.

For the iteration process, we employ a binary search approach to efficiently find an appropriate word count target $n$. Since we observe the real speech time $t$ generally has a positive correlation with $n$, we can find the initial search interval $[n_l, n_r]$ where $n_l$ produces a speech shorter than $t_l$ and $n_r$ produces one longer than $t_r$, making the interval dividing process possible. More detailed descriptions of the process are in Appendix B.2.

## 4 EXPERIMENT

### 4.1 EXPERIMENT AND EVALUATION SETUP

We use the state-of-the-art multi-agent framework **Agent4Debate** (Zhang et al., 2024) with different backbone LLMs as our baseline. It employs a collaborative architecture with four specialized agents: searcher, analyzer, writer, and reviewer agents. It shows better performance in the LLM-based Debatrix debate metric (Liang et al., 2024) and is preferred by human experts in pairwise comparison. It uses the stage-specific prompts from the AIDebater 2024 competition [3] and uses Tavily [4] as a search engine for evidence retrieval.

We use Gemini-2.0-flash (Team et al., 2023) and DeepSeek-V3 (Liu et al., 2024) as the backbone LLM for Agent4Debate and TreeDebater. For a fair comparison, TreeDebater uses the same Tavily APIs for evidence retrieval before the debate and adopts the same stage-specific prompts, adding the necessary modifications to incorporate the tree information for planning, as shown in Appendix G. To ensure that the debate systems follow the time constraint, we add the rough word budget in the stage-specific prompt for both models. After transferring the statement to the audio via OpenAI TTS, we apply a hard cut on the debate audio: the audio will be trimmed from the last sentence before the time limitation. We train two separate LLaMA-based reward models, specifically with the base model being meta-llama/Llama-3.2-3B-Instruct, using the Kialo dataset (Durmus et al., 2019) for $r_s$ and $r_a$. The Kialo dataset consists of claim texts for 741 controversial motions with three categories (Impactful, Medium Impactful, and Not Impactful). During inference, we use the weighted category as the final prediction to provide a more fine-grained score. The other implementation details are in Appendix B.3.

---

[3]http://www.fudan-disc.com/sharedtask/AIDebater24
[4]https://tavily.com

**Stage-level Comparison: Head-to-Head Human Evaluation** We design a controlled head-to-head setting for stage-level comparison. We first have a debate competition between the two debate systems with a random stance assignment. Then we use the debate process before a target stage as the context, and let each debate system generate one version for this target stage. We then let the participant compare the two versions based on the debate process. For example, we keep the same Pro's Opening and ask Agent4Debate and `TreeDebater` to generate the Con's Opening. This head-to-head comparison provides a detailed and focused evaluation of each stage by alleviating the noise introduced by other factors. Specifically, the annotator will provide persuasiveness scores for both versions and also mark which side performs better in each stage. We randomly sample 10 (motion, stance assignment) settings for each stage and get the corresponding debate context, , producing 120 stage-level comparisons. Then we let each debate system generate one statement for the target stage. The audio of the debate context and the two versions of the statement are presented to the participants, where we randomly set the order of these two versions to avoid the potential order bias.

**Debate-level: End-to-End Human Votes** In addition, we follow Oxford-style debating to assess the debater's capability in a full debate. We ask the participants to vote before and after listening to the full 20-minute debate, and provide the persuasiveness score for each stage. We flip the stance assignment and take the average of two competitions for each motion to avoid the potential prior bias on the motion. We randomly sample 4 motions for Gemini and 7 motions for DeepSeek. We conduct two debates on each motion with the original and swapped sides. For each debate, 3 random participants from our recruitment pool are asked to score each stage and vote before and after the debate. The average score and the opinion shift are used to evaluate the debaters' performance in this debate. We take the average of the two flipped stance assignments as the debater's performance on this motion. For example, the opening persuasive score is the average of the opening scores when the debaters acted as the Pro and Con side, respectively. Results are shown in Table 2.

**Recruitment and Quality Control** We recruit audience members from the online research platform Prolific [5]. The evaluation takes around 30 minutes per stage. The compensation is $10 for each case after they complete the evaluation. We recruited 212 participants in total for our debate, and all participants are located in the US when they conduct the evaluation. The screenshot of our evaluation platform is shown in Figure 5 and 6, and the demographic features of our participants are shown in Figure 7. For the stage-level head-to-head evaluation, we calculate the annotators' agreement on the better version. The annotators achieved 60.7% agreement on average, indicating moderate consensus. This agreement indicates a reliable human evaluation result, considering the subjectiveness in debate evaluation. [6]

Before human evaluation, we perform validity checks to assess the quality of the debate competition generated. Only those that follow the correct debate format are used for human evaluation. The results of the valid checks are shown in Appendix E. In general, `TreeDebater` can always generate format-valid and time-valid statements. However, only 77% debates are valid in the Gemini version of Agent4Debate. Besides, Agent4Debate always exceeds the time limitation, especially in the closing stage. We put the motion list (Table 5), persuasiveness score rubric (Figure 5), the recruitment details, and a screenshot of our debate evaluation platform (Figure 6) in Appendix C.

### 4.2 EXPERIMENT RESULTS

**`TreeDebater` has a higher average persuasiveness and win rate in 11/12 stage-level comparison.** As shown in Table 1, the advantage is more significant when we shift from Gemini to DeepSeek. We can find a clearer win in the stage-level win rate. When participants are asked to choose the better version, `TreeDebater` is preferred 1.5x and 2.5x times than the baseline. One reason could be that participants are more likely to choose scores of 3 or 4, leading to a small performance gap in the persuasiveness score.

**`TreeDebater` outperforms the baseline with a 3.5x and 1.3x higher opinion shift win rate in Gemini and DeepSeek.** Table 2 indicates that `TreeDebater` makes the LLM a more persuasive

---

[5]https://www.prolific.com/

[6]In debate-level human evaluation, the winner of the debate is decided based on how many annotators shift towards it, and annotators do not need to agree with each other. Therefore, annotators' agreement is not applicable here.

Table 1: Scalar persuasiveness score and win rate in head-to-head human evaluation. A higher score indicates the statement is more persuasive. Win:Tie:Lose indicates the number of cases that `TreeDebater` wins, gets a tie, or loses in the pairwise comparison. The standard deviation results are put in Table 7 in the appendix because of the length limit.

| Model | Framework | Opening | | Rebuttal | | Closing | | Average |
| --- | --- | --- | --- | --- | --- | --- | --- | --- |
| | | Pro | Con | Pro | Con | Pro | Con | |
| Persuasiveness Score | | | | | | | | |
| Gemini | Agent4Debate | 3.64 | 3.64 | 3.94 | 3.65 | 3.64 | 2.75 | 3.54 |
| | `TreeDebater` | 3.73 | 3.91 | 4.06 | 3.25 | 3.91 | 3.25 | **3.69** |
| DeepSeek | Agent4Debate | 3.45 | 3.18 | 3.73 | 3.55 | 3.45 | 3.45 | 3.47 |
| | `TreeDebater` | 4.27 | 3.36 | 4.18 | 3.73 | 4.27 | 4.27 | **4.01** |
| Win Rate of `TreeDebater` | | | | | | | | |
| Gemini | Win:Tie:Lose | 5:3:3 | 6:1:4 | 2:6:1 | 2:3:5 | 7:1:3 | 5:2:1 | **0.45**:0.27:0.28 |
| DeepSeek | Win:Tie:Lose | 8:2:1 | 4:4:3 | 8:1:2 | 4:4:3 | 3:5:3 | 6:4:1 | **0.50**:0.30:0.20 |

Table 2: End-to-End human evaluation result. The score in each stage indicates how persuasive the statement is. Opinion Shift Win indicates the percentage of votes that shift towards its stance after the debate. We ignore the percentage of Tie here.

| Model | Framework | Opening | Rebuttal | Closing | Opinion Shift Win |
| --- | --- | --- | --- | --- | --- |
| Gemini | Agent4Debate | $2.96_{\pm 0.35}$ | $2.92_{\pm 0.25}$ | $2.98_{\pm 0.29}$ | 0.13 |
| | `TreeDebater` | $\mathbf{3.60}_{\pm 0.27}$ | $\mathbf{3.58}_{\pm 0.25}$ | $\mathbf{3.54}_{\pm 0.29}$ | **0.46** |
| DeepSeek | Agent4Debate | $3.73_{\pm 0.24}$ | $3.81_{\pm 0.15}$ | $\mathbf{3.57}_{\pm 0.23}$ | 0.30 |
| | `TreeDebater` | $\mathbf{3.87}_{\pm 0.33}$ | $\mathbf{3.71}_{\pm 0.28}$ | $3.38_{\pm 0.25}$ | **0.40** |

debater that can change the audience's opinion in the competitive debate. Besides, the debate systems with the DeepSeek backbone have higher persuasiveness scores than Gemini, indicating that DeepSeek has superior capability in argumentation. Moreover, our `TreeDebater` shows a significant improvement in the persuasiveness score at all stages. We find that this is because the poor performance of the Gemini-based baseline in the early stage affects the participants' impression of it and makes them more critical in the following stage.

On the other hand, we find that the audience focuses more on the personal belief in the motion rather than the debate strategies if the two sides have reasonably good performance in the debate. For example, the Agent4Debate baseline already shows an average persuasiveness score of 3 or more. In its pairwise comparison with `TreeDebater`, we find that in the 7 motions we used in our DeepSeek experiments, the Con side always wins in 3 of them and the Pro side always wins in one of them, regardless of the stance assignment. This leads to a comparable persuasiveness score after taking the average of the two flipped assignments, as shown in Table 2. In such a scenario, the head-to-head evaluation provides more insight into the performance difference by focusing more on the debate strategies used under the same debate context.

## 4.3 ANALYSIS

We provide more qualitative and quantitative analysis of `TreeDebater`'s debate performance to investigate how the proposed tree structures help the strategic planning in debate.

Table 3: Ablation Studies on Stage-level Head-to-Head Human Evaluation

| Framework | Opening | Rebuttal | Closing |
| --- | --- | --- | --- |
| `TreeDebater` | 3.50 | 3.50 | 3.75 |
| `TreeDebater` w/o Rehearsal Tree | 3.00 | 3.25 | 3.50 |
| `TreeDebater` w/o Rehearsal & Debater Flow Tree | 3.00 | 3.00 | 3.50 |

**Debate flow trees help `TreeDebater` choose diverse actions, aligning better with human experts' strategies.** We extract the type of debate action used by human debate experts,

`TreeDebater`, and Agent4Debate in the rebuttal statement and plot the distribution in Figure 2. Note that one argument can be identified with multiple action types. For example, if the attack from the opponent is related to the opponent's main claim, then a rebut to this attack can also be viewed as an attack on the opponent's main claim. We consider such a combination of actions as a separate action category. From Figure 2 we can find that `TreeDebater` has more diverse actions in its rebuttal, where attack & rebut, sole attack, and sole reinforce account for a large percentage. This is similar to human experts' strategies in the debate competition: rather than rebutting or attacking every point in the opponent's last statement, they will remind the audience about their earlier claims and shift the focus of the debate to their battlefield. Instead, Agent4Debate is busy attacking and rebutting the latest statement, lacking a long-term action to reinforce its main claims in the opening stage. We also conduct the ablation study by removing the Rehearsal Tree $T_r$ and the Debate Flow Tree $T_d$ in Figure 2. It shows that after removing the flow trees from the debate, `TreeDebater` loses track of the points mentioned in the debate, leading to less diverse actions.

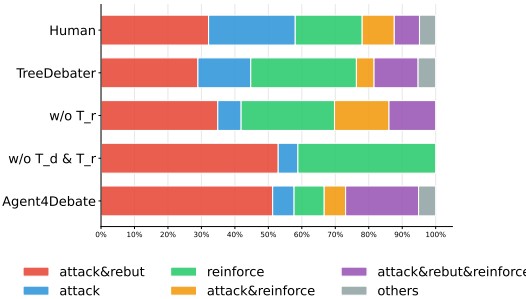

Figure 2: Action distribution in the rebuttal stage. We extract the Debate Flow Tree from human debates and categorize the distribution of actions. The actions are less diverse in the baseline and `TreeDebater` w/o $T_d$.

Figure 3: Percentage of actions that can be found in the Rehearsal Trees. The Rehearsal Trees of both sides contribute to the hit rate.

**Rehearsal trees get `TreeDebater` prepared for the debate by anticipating the opponent's behaviors.** We investigate how many points mentioned in the debate have been anticipated and prepared in the Rehearsal Tree. In Figure 3, we calculate the hit rate of each type of action on the Rehearsal Trees $T_r$ prepared for the debater's side and those $T_{r_{oppo}}$ for the opposite side. One action is hit if there is a node in the Rehearsal Tree that has a similar claim to the target claim of this action, leading to a non-empty retrieval set $R$ in Alg 3 is not empty. Figure 3 shows that more than half of the candidate actions have been anticipated in the Rehearsal Tree, helping `TreeDebater` prepare these actions in advance. It also shows that the Rehearsal Trees from the debater's and the opponent's side both contribute to the preparation by providing different perspectives of the motion. We also find that these *propose* actions have a high hit rate in the opposite Rehearsal Tree. It indicates that the opponent may easily anticipate and prepare for the main claims of the debater's side by treating them as potential attacks on their main claims.

**Qualitative Analysis based on Audience Feedback** The participants are also encouraged to provide optional comments, which gives us more insights into the debate performance. We show several comments in the head-to-head comparison. We categorize the comments into four aspects: *logic flow, audience awareness, evidence, and claim*. In Table 4, we can find that `TreeDebater` outperforms the baseline in the logic flow and has more good points and evidence. This benefits from the flow tracking the Debate Flow Tree and the prepared claims and evidence in Rehearsal Trees. In two lost cases (4 and 5), the participants complain about the statements' emotion, thinking they are too critical. However, the other participants think the emotional tone makes it more convincing. This is because the simulated audience in `TreeDebater` retrieves similar human Debate Flow Trees as references to provide feedback, guiding the statement to have a more human-like tone. Such revision may not favor the audience who prefers an objective statement.

**Fine-grained Ablation Studies for Impact on Persuasiveness** We further conduct the head-to-head evaluation on the ablation of the Rehearsal Tree and Debate Flow Tree. We select two new motions and use Gemini model as the backbone. The persuasiveness scores are shown Table 3. It shows clear performance degradation when removing components, confirming their necessity for better debate quality.

Table 4: Detailed audience feedback. We present several cases for each stage. We put the persuasiveness score at the beginning of the comment. We annotate the different aspects: logic flow in blue, audience awareness in green, evidence in orange, and claims in purple. Win indicates our `TreeDebater` outperforms the baseline.

| ID | Stage | Comments on `TreeDebater` | Comments on Agent4Debate |
|----|-------|---------------------------|--------------------------|
| 1 | Opening (win) | 3: Good flow, adressed the audience properly, could use some stronger evidence, overall moderate. | 2: Flow was disjointed and confusing, but included acceptable points with single good piece of evidence, overall weak. |
| 2 | Opening (win) | 5: I think it is more detailed. I appreciated that he delved deeper into critical thinking skills. | 4: I liked the examples and the statistics he cited though I wish there were references. |
| 3 | Rebuttal (win) | 5: Gave great points and comparisons. Gave examples from studies. | 4: Strong statements. Gave good arguments to the opposition's point. |
| 4 | Rebuttal (lose) | 3: too mean | 4 |
| 5 | Rebuttal (lose) | 2: Picking apart every detail of the other side's argument. | 3 |
| 6 | Closing (win) | 4: Made good points and used really good evidence, and tone was also convincing, so overall strong argument. | 3: part of rebuttal rather than the closing statement, and wasn't as convincing nor did it feel really closing to me. |

## 5 CONCLUSION AND DISCUSSION

Competitive debate is an important proxy for critical thinking, argumentation skills, and the ability to understand and make tackle plans among various candidate actions under the complex dynamic interaction. In this paper, we design a strategic debate system `TreeDebater` to strategically plan its debate actions for time-limited competitive debate. It uses the Rehearsal Tree to anticipate the potential attack and defense for preparation, and uses the Debate Flow tree to track the candidate actions and battlefields. It is facilitated with the speech time controller and the simulated audience feedback to provide revision suggestions for coherent and precise time-controlled statements. The results show that `TreeDebater` can significantly improve the performance of the debate in both head-to-head and end-to-end human evaluation, and elicit more human-like diverse strategies.

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

## A    DEBATE TERMINOLOGY

We introduce the debate terminologies used in this paper. A **motion** in the debate is a statement of opinion or proposition to argue. The assigned **stance** $s$ on the debate motion can be *Pro* or *Con*, indicating supporting or opposing the motion, respectively. A **claim** $c$ is a proposition or assertion about a motion, which should be proved through argumentation. The main claims refer to the most important claims that the debater proposes during the opening stage to support their stance. An **argument** $x$ is a set of reasons or evidence to support a claim, typically consisting of premises that lead to a conclusion. A **statement** is the whole passage presented in one debate stage, including a set of claims, arguments, and supporting evidence [7]. There are several typical **actions** that the debater can take in each stage: *propose* a new claim, *reinforce* its existing claims, *attack* the opponent's claims, and *rebut* the opponent's attack. A **battlefield** refers to a conflict zone in the debate, where two sides contest on a specific point. It includes several rounds of propose attack defense.

## B    ADDITIONAL IMPLEMENTATION DETAILS

### B.1    DEBATE FLOW TREE ALGORITHMS

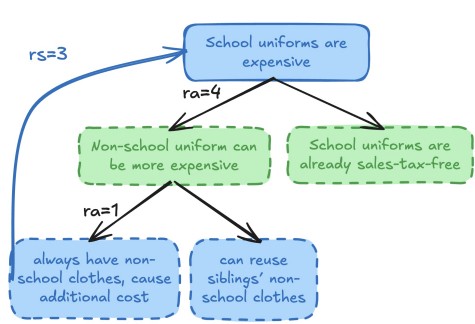

(a) Rehearsal Tree. The root node is the main claim $c$. The blue nodes are from the same side as the root, and the green ones are the potential counter-arguments from the opposite side. $r_s$ is the support score and $r_a$ is the attack score.

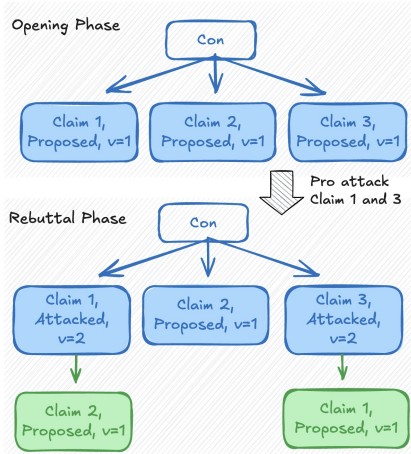

(b) Debate Flow Tree of the Con side. The blue indicates its claims, while the green indicates the claims proposed by its opponent (Pro side) to attack the Con side's claims. $v$ indicates the visit number.

Figure 4: Illustration of Rehearsal Tree (a) and Debate Flow Tree (b).

We demonstrate the Rehearsal Tree and Debate Flow Tree in Figure 4a and 4b. The motion in 4a is '*Should school uniforms be banned*' and the side is Pro. As shown in Figure 4a, the root node is the main claim $c$ proposed for the Pro side of the motion. The blue nodes are the arguments to support the same side of the root (Pro Side), while the green nodes are the counter-arguments that support the opposite side (Con Side).

In Alg. 2 and 3 we provide the pseudo code for updating the Debate Flow Tree with actions tuples and retrieving similar nodes for candidate actions. We use Gemini-text-embedding-4 to get the embedding of the claims. To find the most similar tree node, we traverse all tree nodes and calculate the cos similarity between the claim of the node and the target claim. We set the similarity threshold $\theta$ to 0.8 and return the top one candidate.

---

[7]It is also called a speech (Slonim et al., 2021) or a discourse (Zhang et al., 2016) in some literature.

**Algorithm 2:** Update Debate Flow

**Input:** Debate Tree $T_d$ with root $x^0$,
      Action tuple
      $(action, claim, argu, target)$,
      Similarity threshold $\theta$
**Output:** Updated node

1 **if** $action$ is "propose" **then**
2     Create node $x'$ with $claim$ and $argu$;
3     Add node $x'$ to $x^0$ children ;
4     Modify $x'$ status to "*proposed*" ;
5     **return** $x'$;

6 Find node $x$ similar to $target$ by $\theta$ ;
7 Increase $x$ visit count by $1$ ;
8 **if** $action$ is "reinforce" **then**
9     Update $x$ with $argu$ ;
10     **return** $x$;
11 **if** $action$ is "rebut" or "attack" **then**
12     Create node $x'$ with $claim$ and $argu$;
13     Add node $x'$ to $x$ children ;
14     Modify $x$ status to "*attacked*" ;
15     Modify $x'$ status to "*proposed*" ;
16     **return** $x'$;

**Algorithm 3:** Retrieve from Rehearsal Tree

**Input:** Rehearsal Tree set $\mathcal{T}_R$, Action space $\mathcal{A}$,
      Similarity threshold $\theta$, Look ahead step $k$
**Output:** List of actions with retrieved $\mathcal{A}_R$

1 Initialize action list $\mathcal{A}_R$ with empty ;
2 **foreach** $(action, target) \in \mathcal{A}$ **do**
3     Initialize retrieval set $R$ ;
4     **foreach** $tree \in \mathcal{T}_R$ **do**
5         **if** $action$ is "propose" or "reinforce" **then**
6             Find node $x^l$ similar to $target$ on the *current* side by $\theta$;
7             Add $x^l$ and its $k$-step lookahead strength $f_k(x^l)$ to $R$ if exists ;
8         **else if** $action$ is "attack" or "rebut" **then**
9             Find node $x^l$ similar to $target$ on the *opposite* side by $\theta$;
10             **foreach** $x^{l+1}$ in $x^l$ children **do**
11                 Add $x^{l+1}$ and its $k$-step lookahead strength $f_k(x^{l+1})$ to $R$ if exists ;
12     Add the tuple $action, target, R$ to $\mathcal{A}_R$
13 **return** $\mathcal{A}_R$;

## B.2 SEARCH ALGORITHM IN SPEECH TIME CONTROL

As mentioned in the main text, our goal is to generate a coherent speech within the time range $[t_l, t_r]$ with the chosen actions from the Debate Flow Tree. Note that the number of words is easier to control than the entire speech length for the large language model generation, and the passage word length has a relatively good correlation with the speech length. We can formulate the problem as that we want to call $f(n)$ minimal times to find an output that has speech time in the range $[t_l, t_r]$, where $f$ represents an LLM inference where the speech length of $f(n)$ is correlated to $n$.

Note that if at any time the generated speech time ranges in $[t_l, t_r]$, we can directly finish the whole process. In our implementation, we first try to establish the boundary word counts for binary search $[n_l, n_r]$, where $n_l$ produces a speech shorter than $t_l$ and $n_r$ produces one longer than $t_r$. Specifically, we make an initial query that can either become $l$ or $r$, so that we can find the other endpoint by iteratively doubling $r - l$. With this initial interval determined, we can then iteratively refine this interval through binary search, selecting the midpoint and reducing the interval range by half based on the resulting speech duration.

Note that the effectiveness of this algorithm also depends on how well the base model's generation correlates with the number of words written in the prompt. We observe that while the Gemini output can vary according to the constraint, DeepSeek sometimes seems to disregard the length constraint, making the iterative process slow to stop. We thus set the maximum iteration time as 10 to make sure the algorithm always stops. If the goal is still not accomplished after the maximum time, we will use the latest transcript as the final one.

## B.3 REWARD MODEL TRAINING DETAILS

We train two separate LLaMA-based reward models, specifically with the base model being `meta-llama/Llama-3.2-3B-Instruct`, using the Kialo dataset (Durmus et al., 2019) for $r_s$ and $r_a$. The Kialo dataset consists of claim texts for 741 controversial motions. For each motion, it maintains an argument tree. The root of the tree is the motion, and each node is a claim text with its counter-arguments or subsequent arguments as its children. There are human annotations for the impact score (*Impactful, Medium Impactful, and Not Impactful*) between the node and its parent,

indicating how impactful the node is to support or oppose its parent claim. We preprocess the original dataset to build one dataset with 3691 argument trees for the impact score of the support relation and one dataset with 3676 argument trees for the rebuttal relation and train models for them separately. To make different classes more balanced, we also incorporate a data resampling mechanism to let each class' samples occur multiple times according to the ratio of the maximum class count to the number of samples in this class. We then train the base models in the classification task for 3 epochs. Finally, our trained reward model gets 0.67 accuracy in predicting the support impactfulness and 0.72 on the rebuttal impactfulness. During finetuning, we formulate the original data point with the following prompt and let the llama predict the impact category. We view it as a classification task and use LlamaForSequenceClassification as the backbone architecture and the cross-entropy as the loss function. The instruction format is shown below.

---

**Instruction format of LLaMA-based reward models**

```
You are given a chain of arguments, each one supporting or attacking
the previous one.
The previous arguments are: [context]
The second last one is: [claim 1]
The last one is: [claim 2]
Now you need to determine the impact of the last one on the second
last one, given their relationship [support/attack]. Output only a
number among 0, 1, or 2 in your response. 0 means not impactful; 1
means medium impactful; 2 means impactful.
```

---

After applying $r_a$ and $r_s$ to get the strength score by Eqn. 2, the Rehearsal Tree will look like:

---

**One example in Rehearsal Tree**

```
Level-0 Root Claim: "claim": "Removing the debt ceiling benefits
future generations.", Scores: Support Score: 1.6, Strength: 0.9
Level-1 Opponent's Attack: "claim": "The debt ceiling, while
imperfect, compels fiscal responsibility, safeguarding future
generations from unsustainable debt burdens.", Scores: Attack Score:
1.3, Strength: 0.8
Level-2 Your Rebuttal: "claim": "The debt ceiling does not ensure
fiscal responsibility; it merely invites fiscal crises.", Scores:
Attack Score: 1.4, Support Score: 1.6, Strength: 0.6
Level-3 Opponent's Attack: "claim": "The debt ceiling's
'manufactured crises' are preferable to unchecked spending, which is
a greater danger.", Scores: Attack Score: 0.8, Support Score: 0.8,
Strength: 0.8
Level-3 Opponent's Attack: "claim": "Our budgeting process
requires improvement, but eliminating the debt ceiling is not the
right solution.", Scores: Attack Score: 1.0, Support Score: 0.8,
Strength: 0.9
Level-3 Opponent's Attack: "claim": "The debt ceiling does not
need to be a 'political weapon' and can be reformed to work more
effectively, rather than eliminated.", Scores: Attack Score: 1.2,
Support Score: 1.0, Strength: 1.1
```

---

## C  HUMAN EVALUATION DETAILS

**Motion List** We collect 52 debate motions from different sources, including the recent hot topics in the debate website OpentoDebate[8] and the opinion section in the New York Times, the motions used in previous debate systems (Slonim et al., 2021; Zhang et al., 2024), and the persuasiveness dataset released by Anthropic (Durmus et al., 2024). It covers various domains such as Economics, Finance, Health, Science, Culture, and etc. We then asked two human expert debaters to annotate how

---

[8]https://opentodebate.org/

polarized the motions are based on a 1-5 scale. Finally, we keep 13 motions that are less polarized. We list motions and their sources in Table 5. Note that these topics are only used as a debate motion. We do not use the background materials or debate transcripts in their sources.

Table 5: Motion List

| ID | Motion | Domain | Source |
|---|---|---|---|
| 1 | Congress should abolish the debt ceiling | Economics | OpentoDebate |
| 2 | Labor unions are beneficial to economic growth | Finance | OpentoDebate |
| 3 | The United States should implement a central bank digital currency | Finance | OpentoDebate |
| 4 | Processed foods should play a larger role in sustainable food systems | Health | OpentoDebate |
| 5 | AI will lead to the decline of human creative arts | Science | OpentoDebate |
| 6 | It is time to welcome an A.I. Tutor in the classroom | Technology | New York Times |
| 7 | Dating Expenses Should Be Shared Equally Between Partners | Culture | New York Times |
| 8 | Mandatory wage transparency laws should be implemented to address the gender wage gap | Economics | OpentoDebate |
| 9 | Artists should be free to borrow from cultures other than their own | Culture | OpentoDebate |
| 10 | If health care is a scarce resource, government should step in to ration care, deciding whose life is worth saving | Health | Oxford Dataset |
| 11 | We should ban certain inappropriate books (like sex violence drug use) in school | Education | OpentoDebate |
| 12 | Developed countries should impose a fat tax. | Health | Agent4Debate |
| 13 | Pursuing a four-year college degree remains beneficial for young adults in today's society | Education | New York Times |

**Recruitment** We recruit audience members from the online research platform Prolific [9]. The evaluation takes around 30 minutes per stage. The compensation is $10 for each case after they complete the evaluation, which is higher than the minimum wage of $7.25 per hour in the United States. The participants should be older than 18 years old, be fluent in English, and have achieved a high school degree. We have two attention check questions during the evaluation to control the quality. Two screening questions are : (i) a multiple-choice question to choose the main claim proposed by a specific side, with multiple confusing options that are difficult to distinguish; (ii) a free-form QA question to summarize the key idea of one side. We filter out responses that fail the screening questions (about 10%). One example is shown below. The screenshot of our evaluation platform is shown in Figure 5 and 6. The demographic survey results of our participants are shown in Figure 7.

---

**Two screening questions**

```
Q1:  Which claim is proposed by For side as its first main claim
during the opening statement?  If multiple options apply, please
choose the best one.
(A) The debt ceiling creates unnecessary political crises that harm
the economy.
(B) Effective Altruism's metrics-driven approach overlooks crucial
local contexts that determine real impact
(C) The debt ceiling undermines the full faith and credit of the
United States.
(D) The debt ceiling encourages bipartisan negotiation on fiscal
policy.
(E) None of the above

Q2:  Which is the main battlefield / conflict / question mentioned
by For side in its closing statement?  *
```

---

**Rubrics for Evaluation** We provide the audio and optional transcripts for each statement. Participants are asked to listen to the audio and provide the persuasiveness score for each audio. In the head-to-

---

[9]https://www.prolific.com/

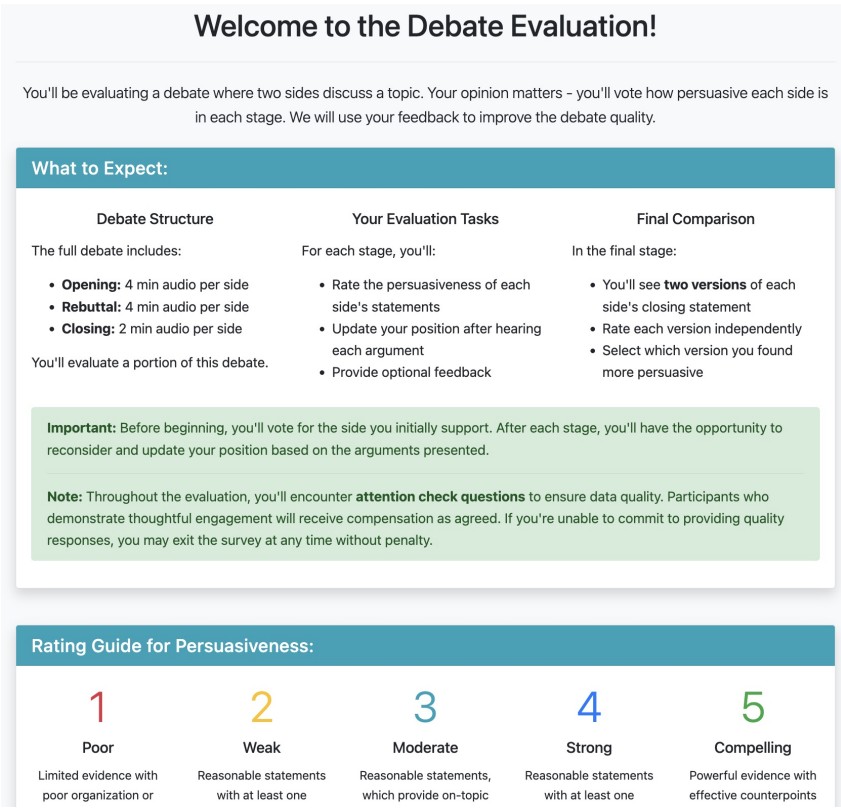

Figure 5: The screenshot of the instruction given to the participants in our human evaluation platform

head evaluation, they are also asked to choose the preferred version. In the end-to-end evaluation, they are asked about the attitude change after each stage. They can read the optional transcript and provide optional comments for each stage.

Participants are asked to provide a 1-5 persuasiveness score for each stage/version based on the following rubric:

- **Poor (1)**: Limited evidence with poor organization or fundamental logic flaws. Disengage with no audience awareness.
- **Weak (2)**: Reasonable statements with at least one noticeable weakness.
- **Moderate (3)**: Reasonable statements, which provide on-motion evidence with logical flow and balanced emotional tone showing basic audience awareness
- **Strong (4)**: Reasonable statements with at least one impressive shining points.
- **Compelling (5)**: Powerful evidence with effective counterpoints and creates a deep connection with audience.

**Experiment cost** We conducted 66 head-to-head comparisons and 36 end-to-end comparisons for each backbone model. We further hire 6 human debate experts to provide detailed feedback via a 1-hour one-on-one interview. The human debate experts have participated in at least 10 debate competitions. We provide a $30 Amazon gift card as compensation for each case they evaluate. We collect 13 expert feedback in total. The total cost for human evaluation is about $3k, including the platform fee.

**IRB details** We provide detailed instruction to participants in our evaluation platform, which is shown in Figure 5. All participants should complete the consent form before conducting the study. Compensation is described above.

Motion: Labor Unions Are Beneficial To Economic Growth

**Question 1: Pre-Vote Stage**

**Please choose your attitude towards the given motion before the debate begins.** *

○ Highly Against
○ Slightly Against
◉ Neutral
○ Slightly Support
○ Highly Support

**End of Pre-Vote Stage Comparison**

**Question 2: Opening Stage**

**For Side**   ▶ 0:00 / 3:19 ━━━━━━━━━━━━━━━━━━━━━━━━━━━━  🔊 ⋮

▶ (Optional) **For** - Transcript

How persuasive are these arguments in supporting **For** side? Rate the performance based on the following principle. *

○ **Poor**: Limited evidence with poor organization or fundamental logic flaws. Disengage with no audience awareness.
○ **Weak**: Reasonable statements with at least one noticeable weakness.
○ **Moderate**: Reasonable statements, which provide on-topic evidence with logical flow and balanced emotional tone showing basic audience awareness
○ **Strong**: Reasonable statements with at least one impressive shining points.
○ **Compelling**: Powerful evidence with effective counterpoints and create deep connection with audience.

(Optional) Additional comments for **For** side:

|  |
|---|

| **Output A - Against Side** | **Output B - Against Side** |
|---|---|
| ▶ 0:00 / 3:50 ━━━━━━━━━ 🔊 ⋮ | ▶ 0:00 / 3:36 ━━━━━━━━━ 🔊 ⋮ |
| ▶ (Optional) **Against** - Transcript A | ▶ (Optional) **Against** Transcript B |
| How persuasive is **Output A** in supporting **Against** side? Rate the performance based on the following principle. * | How persuasive is **Output B** in supporting **Against** side? Rate the performance based on the following principle. * |
| ○ **Poor**: Limited evidence with poor organization or fundamental logic flaws. Disengage with no audience awareness. | ○ **Poor**: Limited evidence with poor organization or fundamental logic flaws. Disengage with no audience awareness. |

Figure 6: The screenshot of the head-to-head evaluation in our human evaluation platform. We provide the audio and the optional transcripts for each statement.

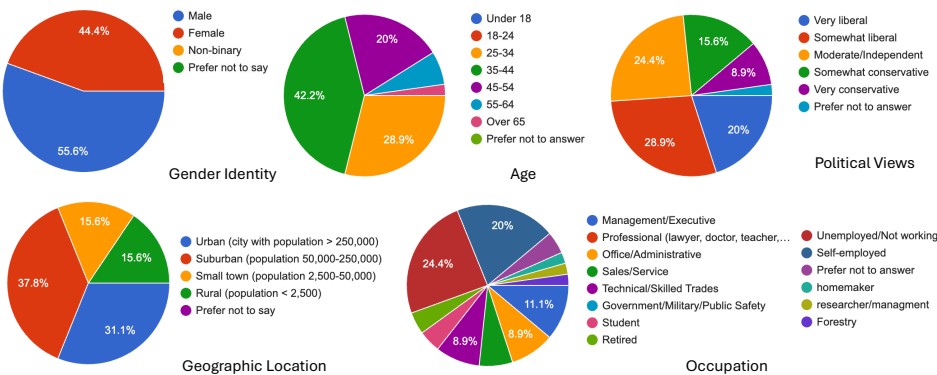

Figure 7: Demographic Survey of our participants

## D    HUMAN DEBATE FLOW TREE CORPUS

We create the human Debate Flow Tree corpus from the human debate dataset PanelBench (Liang et al., 2024). PanelBench is built from the online debate platforms DebateArt and world-class competitive debate competitions, BP-Competition. The number of debate is 122. We follow the MIT license of PanelBench. We first reorganize the original debate transcript into the opening, rebuttal, and closing statements. Then, for each statement, we extract the debate action tuples (action, claim, argument, target claim) via the backbone LLM and build the Debate Flow Tree based on Alg. 2 for each debate. These human Debate Flow Trees will then be retrieved during the debate to provide feedback for `TreeDebater`. There is no overlap between the motions used in our experiments with the human Debate Flow Tree corpus.

To endow `TreeDebater` with sophisticated, human-like debating capabilities, we incorporate the human debate tree into the pipeline. As highlighted in Section 1, human experts intuitively employ tree-like structures to organize potential arguments and to meticulously track the evolving flow of a debate. Our human debate feedback mechanism operationalizes this human-centric approach. Specifically, during the debate, we reformulate the current Debate Flow Tree into a tree-like string. Then we employ semantic search based on the embeddings of the debate flow tree string. One of the state-of-the-art embedding models Gemini-text-embedding-4 is used to get the embedding of the tree-like string of the current debate flow tree and of trees in human debate flow tree corpus. The top 1 human debate flow tree is retrieved with a similar threshold of 0.8. We add the retrieval to the instruction for simulated audience to let it aware of how human debate on similar topics.

## E    VALIDITY CHECK

We first investigate whether these two debate systems can generate a debate statement that follows the correct format and the time constraint. A statement is viewed as invalid if it only lists key points without concrete arguments, includes intermediate thoughts such as '*I will provide feedback on...*', or even misidentifies of its stance. A statement with valid time indicates that the audio version satisfies the time constraint before the hard cut.

As shown in Table 6, our `TreeDebater` can always generate valid statement. Agent4Debate with Gemini has difficulty generating format-valid statements. We find that it often takes the feedback of the reviewer agents as the final debate statement, or explicitly mentions that '*as suggested by the reviewer*' in their statement, which is a common issue in the multi-agent system. This can be mitigated with a better backbone LLM.

However, even with a more powerful DeepSeek , Agent4Debate still suffers from the time constraint. It is difficult for it to generate statements with the required word number, resulting in a low time validity in all stages. This is more severe in the closing stage, where the debate system is required to summarize the full debate with a stricter 2-minute limitation. Instead, our `TreeDebater` benefits from the iterative revision based on the speech time controller and can guarantee the audio time of the statement to be in the required range. For human evaluation, we filter out the cases with invalid format and use the hard cut to ensure the time validity.

Table 6: Percentage of valid debate statements. Format Validity is the percentage of the debate competitions where all statements have the correct format. Time validity is the percentage of the statements that meet the time constraint before the hard cut.

| Model | Framework | Format Validity | Time Validity | | |
|---|---|---|---|---|---|
| | | | Opening | Rebuttal | Closing |
| Gemini | Agent4Debate | 77.0% | 63.5% | 75.0% | 13.5% |
| | TreeDebater | 100.0% | 100.0% | 100.0% | 100.0% |
| DeepSeek | Agent4Debate | 100.0% | 98.1% | 94.2% | 5.8% |
| | TreeDebater | 100.0% | 100.0% | 100.0% | 100.0% |

Table 7: Standard deviation in head-to-head evaluation.

| Model | Framework | Opening | | Rebuttal | | Closing | |
|---|---|---|---|---|---|---|---|
| | | For | Against | For | Against | For | Against |
| Gemini | Agent4Debate | 0.92 | 1.29 | 0.57 | 1.29 | 0.67 | 0.69 |
| | TreeDebater | 0.90 | 0.94 | 0.82 | 1.51 | 1.14 | 1.25 |
| DeepSeek | Agent4Debate | 0.93 | 0.87 | 0.65 | 0.93 | 0.94 | 0.90 |
| | TreeDebater | 1.27 | 1.29 | 0.98 | 0.90 | 0.94 | 0.50 |

Table 8: Standard deviation in end-to-end evaluation.

| Model | Framework | Opening | Rebuttal | Closing |
|---|---|---|---|---|
| Gemini | Agent4Debate | 0.35 | 0.25 | 0.29 |
| | TreeDebater | 0.27 | 0.25 | 0.29 |
| DeepSeek | Agent4Debate | 0.24 | 0.15 | 0.23 |
| | TreeDebater | 0.33 | 0.28 | 0.25 |

## F   ADDITIONAL EXPERIMENTAL RESULTS

The standard deviation of Table 1 is shown in Table 7.

We conducted 10 additional evaluations for each stage, providing a more robust statistical foundation. As shown below, our method consistently outperforms baselines across 5 head-to-head stages in stage-level win rate, with comparable performance in the remaining 1 stage. Among the 5 stages, 3 of them (Opening Pro, Rebuttal Pro, and Closing Con) showed significant improvements (95% CI, $p<0.05$). Critically, our method shows significant overall performance (95% CI, $p<0.05$). Notably, in debate settings, it is almost unlikely for one debater to win all stages without ties, even for human debate experts. Considering the overall performance, the consistently better win rate with significant overall gains represents strong empirical evidence of our method's effectiveness. We will add this statistical reporting, along with the standard deviations, to our revised manuscript.

Table 9: Statistical Significance. Stages with * indicate statistical significance, which are demonstrated by the positive confidence interval and p-value < 0.05.

| | Our Win Rate | Confidence Interval (CI 95%) | P-value |
|---|---|---|---|
| Opening Pro* | 88.90% | (0.225, 1.047) | 0.002 |
| Opening Con | 57.10% | / | >0.05 |
| Rebuttal Pro* | 70.00% | (0.054, 0.612) | 0.019 |
| Rebuttal Con | 57.10% | / | >0.05 |
| Closing Pro | 50.00% | / | >0.05 |
| Closing Con* | 85.70% | (0.104, 0.646) | 0.006 |
| Overall* | 71.73% | (0.138, 0.459) | 0.0002 |

## G   MAIN PROMPTS

Each stage of speech generation will utilize the Debate Flow Tree, where we have prompts in Table 12 for the opening stage, Table 13 for the rebuttal stage, and Table 14 for the closing stage. Regarding the Rehearsal Tree, we have Table 11 for the selection of the main claim based on the Rehearsal Tree.

## H   EXAMPLE OF HUMAN DEBATE BASED FEEDBACK

Table 15 shows an example feedback from the human debate based feedback mechanism introduced in Section 3.4.

| Main Claim Generation Prompt |
| --- |
| ## Task: Generate Strategic Counter-Arguments
You are participating in a formal debate on the motion: {motion}
Your position: {act} the motion

## Your Objective
Generate num persuasive counter-arguments that:

## Context
Previous debate exchanges:
{history} |

Table 10: Main Claim Generation Prompt.

| Main Claim Selection Prompt |
| --- |
| ## Task: Select Persuasive Claims for Debate
You are participating in a formal debate on the topic: {motion}. Your position is {side}.
Select most persuasive claims from the provided options, using the debate tree information.

## Simulated Debate Flow Tree Structure
Each claim has a simulated debate flow tree that simluate the potential back-and-forth between you and your opponent under this claim:
* Level-0: The root claim (potential main claim for selection)
* Level-1: Your opponent's rebuttal to the root claim
* Level-2: Your defense against the opponent's rebuttal

## Input
**Definition of the debate topic**: {definition}
**Simulated Debate Flow Tree for each claim**: {tree}
**Opponent's opening statement**: {context}
**Claims to select from (All Level-0 claims)**: {claims}

## Output
Provide results in JSON format with three fields under the key of *selection*:
**claims**: a list of your selected claims. Each claim is a string. It usually contains 3 *very different claims* from non-overlapping perspectives.
**framework**: String describing the logical structure connecting these claims
**explanation**: String explaining how this framework support your stance and rebut the opponent's opening statement (if provided) |

Table 11: Main Claim Selection Prompt.

---

**Opening Stage Generation Prompt**

---

The debate topic is: {motion}. Your side is to {act} this topic . Now it comes the opening phase. A complete opening statement should include definitions, judging criteria, and arguments.

## Rules
- Ensure your language is fluent and natural, indistinguishable from human writing. Make sure your debate script is complete, including definitions, judging criteria, and arguments.
- When citing data and academic research, provide sources within the context and avoid using information not present in the provided materials. Ensure your arguments are supported by data and academic evidence.
- When citing data, **using specific figures** instead of just descriptive language will make your argument more persuasive.
- When citing data and academic research, **don't just** list the information, **but also explain** how it supports your point.
- When choosing evidence, numeric data is preferred over qualitative descriptions.
- List your references in a standard format in a Reference section, making sure each source has a clear number and url information.
- Renumber the original citations to [1],[2], ... and etc. List full references in **Reference** section. This section should start with **Reference**. Use Chicago style for the reference list and DO NOT include the web link or url information.

## Debate Flow Tree Structure
You are given two debate trees that model the back-and-forth between you and your opponent. Each node contains:
* Data: The specific claims and arguments
* Visit Count: Number of times addressed in debate
* Status: 'proposed' (new), 'attacked' (challenged), or 'solved' (resolved)

## Input Information
Debate flow trees with node data:
**Your Tree**: {tree} **Opponent's Tree**: {oppo_tree} **Your Main Claims**: {claims} **Definition**: {definition}

## Battlefields
Allocate time to the most important battlefields first. Present each battlefield as a complete unit.
**Battlefield Importance**: {high/medium/low}
**Battlefield**: {description of the battlefield}
**Battlefield Rationale**: {thoughts of this battlefield}
**Actions**: {$\mathcal{A}_R$ from Alg 3}

## Output with the format
**Opening Plan**: Allocate your word budget and explain your rationale. Briefly mention one or two rhetorical techniques and logical fallacies to discuss. Ensure the total is {n_words} words.
**Statement**: Generate an opening statement of {n_words} words in total, with no additional text

---

Table 12: Opening Stage Generation Prompt. Instructions marked with blue are borrowed from Agent4Debate.

---

**Rebuttal Stage Generation Prompt**

---

Now it comes the rebuttal phase, where you respond to your opponent. The debate topic is: {motion}.
You side is to {act} this topic .
You should stand firm on your side ({act} the topic) and attack the opponent's weak points.

## Knowledge
### Structure of a Rebuttal
A complete rebuttal should consist of multiple points, with each point containing four parts:
- **Lead-in:** Introduce the opponent's argument, evidence, reasoning, etc., that you will be refuting.
- **Explanation:** Briefly explain the opponent's argument to ensure clarity.
- **Rebuttal:** This is the core of your point. Utilize rebuttal techniques to directly challenge the opponent's claim.
- **Impact:** Concisely summarize the impact of your rebuttal and how it benefits your side.
Note: Typically, the lead-in and explanation are combined into one sentence. The rebuttal is the most crucial part, and the impact summarizes its effect.

### Rebuttal Techniques
- **Pointing out logical fallacies:** Identify errors in the opponent's reasoning, such as reversing cause and effect, equivocation (shifting the meaning of a key term), straw man arguments, circular reasoning, or tautology (repeating the same idea in different words).
- **Pointing out factual errors:** Highlight inaccuracies or weaknesses in the opponent's evidence, such as insufficient data, incorrect facts, or biased sources.
- **Pointing out error logic:** Identify flawed logic underlying opponent's framework.
- **Leveling the playing field:** This technique aims to neutralize the opponent's advantage or minimize the perceived harm of your side's position by demonstrating that both sides share the same issue or benefit.
- **Acknowledging and countering:** Start by acknowledging a valid point made by your opponent before explaining why your position still offers a better solution.

## Debate Flow Tree Structure
You are given two debate trees that model the back-and-forth between you and your opponent. Each node contains:
* Data: The specific claims and arguments
* Visit Count: Number of times addressed in debate
* Status: 'proposed' (new), 'attacked' (challenged), or 'solved' (resolved)

## Input Information
Debate flow trees with node data:
**Your Tree**: {tree} **Opponent's Tree**: {oppo_tree}

## Battlefields
Allocate time to the most important battlefields first. Present each battlefield as a complete unit.
**Battlefield Importance**: {high/medium/low}
**Battlefield**: {description of the battlefield}
**Battlefield Rationale**: {thoughts of this battlefield}
**Actions**: {$\mathcal{A}_R$ from Alg 3}

## Output with the format:
**Rebuttal Plan**: First, allocate words for the overview of the rebuttal. Then, allocate the rest of the word budget among the battlefields. Explain your rationale. Briefly mention one or two rhetorical techniques to use and logical fallacies to discuss. Make sure the total words is {n_words}.
**Statement**: After the rebuttal plan, generate a rebuttal statement of {n_words} words in total, do not include any other text

---

Table 13: Rebuttal Stage Generation Prompt.

**Closing Stage Generation Prompt**

Now it comes the closing statement, where you summarize your key points and reaffirm your position ({act} the topic) . Your position is to {act} the topic. The opponent is to {counter_act} the topic.

## Primary Objectives of a Closing Statement
- Convince the judges that your team won more battlegrounds.
- Demonstrate your team's strengths and the opponent's weaknesses within each battleground based on the clash outcomes.

## Rules - Ensure your writing is fluent, natural, and indistinguishable from human writing.
- Avoid empty appeals to values. Remember, value appeals should connect back to the topic and your stance.
- When citing data or theories, provide sources. Do not introduce new information or data in the closing statement.
- Avoid repeating arguments from previous speeches. Instead of mechanically listing points, focus on deepening your arguments and ensuring logical coherence.
- This is a closing statement, not the time for new arguments. Prioritize depth over breadth.
- Base your arguments on the identified battlegrounds and clashes.

## Tree Structures
Two debate flow trees track the exchange of arguments. Each node contains:
* Data: The specific claims and arguments
* Visit Count: Number of times addressed in debate
* Status: 'proposed' (new), 'attacked' (challenged), or 'solved' (resolved)

## Input Information
Debate flow trees with node data:
**Your Tree**: {tree} **Opponent's Tree**: {oppo_tree}

## Output with the format:
**Closing Plan**: Allocate your word budget based on the debate flow trees and explain your rationale. Make sure the total words is {n_words}.
**Statement**: after the closing plan, generate a closing statement of {n_words} words in total, do not include any other text

Table 14: Closing Stage Generation Prompt.

**Example of Human Debated Based Feedback**

## Comprehensive Analysis
**Core Message Clarity**: The core message is clear: when healthcare is scarce, the government should ration it. The definitions provided are helpful in framing the debate. The logical flow, presenting the rationale as combating discrimination, ensuring public accountability, and supporting medical innovation, is easy to follow.
**Engagement Impact**: The opening does a decent job of setting the stage, but it lacks emotional resonance. The water analogy is helpful but could be more vivid. The discussion of societal biases and ethical considerations is likely to resonate with the audience. The opening mainly targets the general audience.
**Evidence Presentation**: The statement primarily relies on logical arguments rather than specific evidence. While the points are well-articulated, the lack of data, case studies, or expert opinions weakens the overall persuasiveness.
**Persuasive Elements**: The argument has practical relevance, particularly given concerns about healthcare costs and access. The emotional resonance could be stronger by highlighting the potential consequences of *not* having government intervention, making the necessity of their position all the more relevant. The call to action isn't explicit, but the implication is that the audience should support government intervention in healthcare rationing.

## Critical Issues and Minimal Revision Suggestions
**1. Issue: Lack of concrete evidence to support the claims.**
Impact on Audience: Weakens the persuasiveness and leaves the audience wondering if these are just theoretical arguments.
Minimal Revision Suggestion: Include statistics or brief examples illustrating the problem of discrimination in healthcare, the failures of private companies in equitable allocation, or the benefits of government-funded medical research.
**2. Issue: Limited emotional resonance.**
Impact on Audience: Fails to fully engage the audience on an emotional level, making the argument less memorable and impactful.
Minimal Revision Suggestion: Add a brief, compelling anecdote or hypothetical scenario that illustrates the potential negative consequences of *not* having government intervention in healthcare rationing, emphasizing the human cost of inequitable allocation.
**3. Issue: The judging criterion justification can be strengthened**
Impact on Audience: The audience may have different perspective on what would be the ideal judging criterion, and it will be ideal to persuade them on this perspective.
Minimal Revision Suggestion: Briefly explain *why* minimizing discrimination and maximizing overall population health should be the criteria used to evaluate the debate, connecting it to fundamental values of fairness and societal well-being. For example, "By focusing on minimizing discrimination and maximizing overall population health, we can ensure a system that is both just and effective in saving lives when resources are limited."
**4. Issue: Lack of clarity on how the government will decide whose life is worth saving.**
Impact on Audience: Raises concerns about potential biases, ethical dilemmas, and the practical implementation of rationing.
Minimal Revision Suggestion: Briefly mention the intention to propose transparent, objective criteria for rationing decisions, referencing examples of existing frameworks (e.g., age, severity of illness, likelihood of survival) that could be considered. For example, "We propose using transparent and objective criteria, such as age, severity of illness, and likelihood of survival, to guide rationing decisions."

Table 15: Example of Human Debated-Based Feedback.

---

**Simulated Audience Feedback Prompt**

---

## Your Task
You are a panel of debate audience members to provide comprehensive feedback on how the statement impacts and persuades a general audience.

### Audience Panel Composition
- General public with varied educational backgrounds
- Students and educators from different fields
- Professionals interested in policy and social issues

### Evaluation Dimensions
1. **Core Message Clarity**
2. **Engagement Impact**
3. **Evidence Presentation**
4. **Persuasive Elements**

### Guidelines
- Evaluate all dimensions thoroughly
- Identify the most significant barriers to audience understanding in the {stage} statement
- Consider which issues could be addressed with minimal revisions on the {stage} statement
- Focus on high-impact, low-disruption improvements

## Retrieval Information
Here are debate flow trees and action allocations from human debates.
Use the structure and allocation strategy to provide better feedback.

{retrieval debate flow tree}

### Input Information
**Debate Topic**: {motion}
**History of the debate**: {history}
**Current {side}'s {stage} Statement to be evaluated**: {statement}

### Output Format
[Comprehensive Analysis]
Core Message Clarity:
Engagement Impact:
Evidence Presentation:
Persuasive Elements:
[Critical Issues and Minimal Revision Suggestions]
Issue:
Impact on Audience:
Minimal Revision Suggestion:

---

Table 16: Simulated audience feedback prompt.

