# OpenReview forum: "Strategic Planning and Rationalizing on Trees Make LLMs Better Debaters"
_ICLR.cc/2026/Conference — ICLR 2026 Poster_

### Official Review · Reviewer_hgPJ · 2025-10-30

**Soundness:** 3
**Presentation:** 2
**Contribution:** 2
**Rating:** 4
**Confidence:** 4

**Summary:**

This paper addresses the task of computational argumentation and presents TreeDebater, a debate framework for LLMs that structures the debate process through two explicit tree-based representations. The Rehearsal Tree enables the model to anticipate potential attacks and defenses prior to the debate, and the Debate Flow Tree dynamically captures the evolving argumentative structure and state throughout the interaction. Additionally, a time control module is introduced to simulate real-world debate conditions by enforcing strict timing constraints. The framework is evaluated at both the stage and debate levels using an Oxford-style debate format, and human evaluations demonstrate that TreeDebater consistently outperforms the Agent4Debate multi-agent framework across multiple aspects.

**Strengths:**

- The paper addresses an important challenge in enabling LLMs to participate in realistic debates. Instead of focusing on static argument generation, this study simulates an interactive debate process under time constraints, making it closer to real-world settings.
- The introduction of the Rehearsal Tree and Debate Flow Tree is well-motivated and inspired by human debate strategies, effectively structuring anticipation and response during the debate.
- The evaluation is fairly comprehensive, covering both head-to-head and end-to-end comparisons to thoroughly assess the system’s performance.
- Table 3 also provides user feedbacks, which may bring insights for future research.

**Weaknesses:**

- The core idea centers on applying tree-based planning and reasoning for argument generation, with the main contribution being the task-specific adaptation of this structure. However, multi-step agent interactions and tree-based debate frameworks are already common strategies for enhancing LLM reasoning, which makes the overall novelty of the work somewhat limited.
- While Figure 2 presents the distribution of action types, the paper lacks fine-grained ablation studies for key components (e.g., simulated audience feedback, speech time controller, retrieval quality). This makes it difficult to isolate and understand the contribution of each module.
- The comparisons are limited to Agent4Debate, without inclusion of other LLM-based argument generation systems. Additionally, the use of powerful base models such as Gemini and DeepSeek raises questions about generalizability: how would the framework perform with smaller models (e.g., 7B parameters) as the backbone LLM?
- I appreciate the inclusion of human evaluations. However, important details are missing: How many annotators assessed each sample? What is the inter-annotator agreement, given the subjective nature of argument evaluation? Moreover, since the generated arguments may vary in claim or stance each time, how is bias controlled to ensure fair evaluation across different runs? (e.g., A reader may favor a model outputs mainly because of the claim/opinion itself rather that the quality of the outputs)
- According to line 798, it appears that 52 debates are generated for the experiments. If correct, this detail should be included in the main paper as this is an important detail. Furthermore, it would strengthen the analysis to include a diversity measure of debate topics to assess the model’s robustness across different propositions.

**Questions:**

Q1. From Figure 4, it appears that each node’s claim is typically concise and often limited to a single sentence. These nodes seem to represent subtopics or subclaims that illustrate the progression from the main claim to supporting points. However, in real debates, intermediate arguments are usually more complex, involving actions such as attacking a premise from the opposing side or providing detailed reasoning to defend an opinion, especially in an interactive debate setting. Therefore, this tree structure might struggle to fully capture the dynamic and intricate reasoning processes that characterize authentic debates. I would like to hear the authors’ comments on this point.

**Details Of Ethics Concerns:**

Debate or argument generation is a subjective task and may introduce biased opinions or harmful contents, yet no ethics statements are provided in the paper.

---

> ### Author Response · Authors · 2025-11-21
>
> We sincerely thank the reviewer for acknowledging the value of our two human-inspired debate trees and human evaluation methodology. We deeply appreciate their insightful suggestions, which have helped us strengthen the manuscript (highlighted in green). We address the remaining concerns below.
>
> **W1: Multi-step agent interactions and tree-based debate frameworks are already common strategies for enhancing LLM reasoning**
>
> A1: Previous tree-based reasoning work (Tree-of-Thought) and multi-agent debate enhance LLM reasoning by finding ground-truth solutions. Our work addresses a fundamentally different problem: improving persuasiveness under opponent attacks when no ground-truth exists.
>
> Competitive debate poses unique challenges in one’s reasoning and planning capability, especially in a time-constraint scenario:
>
> (i) strategic choices about which points to state.
>
> (ii) the back-and-forth interaction between one and its opponent
>
> To address these unique challenges, we
>
> - Anticipate the opponent’s behavior in the Rehearsal Tree and calculate the claim’s strength score with a minimax-like method to consider the potential attack and defense
> - Model the own and opponent’s real-time behaviors in the Debate Flow Tree to select the appropriate debate action to take
> - A speech time controller with revision to meet the time constraint
>
> These can be summarized as below:
>
> | Aspect | Previous reasoning work such as ToT / Multi-agent Debate | Ours |
> | --- | --- | --- |
> | Goal | Find correct solutions | Convince others of assigned opinion within word limits |
> | Challenges  | Complex reasoning steps | Strategic planning to identify effective points and allocate words |
> | Reasoning steps | Internal thoughts only | Own and opponent's behaviors |
> | Output | Unlimited length | Limited length |
> | Evaluation | Correctness | Persuasiveness |
> | Ground Truth | Yes | No |
>
> This represents a shift from single-agent problem-solving to interactive persuasion without ground truth, requiring fundamentally different planning and reasoning approaches.
>
> ---
>
> **W2: Lacks fine-grained ablation studies for key components**
>
> A2: We conducted head-to-head ablation studies on two new motions using Gemini:
>
> | Framework | Opening | Rebuttal | Closing |
> | --- | --- | --- | --- |
> | TreeDebater | 3.50 | 3.50 | 3.75 |
> | TreeDebater w/o Rehearsal Tree | 3.00 | 3.25 | 3.50 |
> | TreeDebater w/o Rehearsal & Debater Flow Tree | 3.00 | 3.00 | 3.50 |
>
> Results show clear performance degradation when removing components, confirming their necessity for better debate quality.
>
> The simulated audience provides feedback on **message clarity, engagement, evidence quality, and persuasiveness** by comparing statements against real human debate (Section 3.4). This feedback helps TreeDebater refine the presentation. Qualitative analysis (Table 4) shows refinements improve audience awareness and emotional tone, making arguments more convincing. See Table 15 (Page 26) for an example.
>
> ---
>
> **W3: No other LLM-based argument generation systems. How the framework perform with smaller models (e.g., 7B parameters) as the backbone LLM?**
>
> A3: We focus on LLM agents conducting full competitive debates, where the key challenge is **planning multiple arguments to maximize persuasiveness**. Other LLM-based argument generation systems enhance single arguments but cannot directly apply to competitive debate settings.
>
> We compare primarily with Agent4Debate because: (i) it outperforms vanilla LLM-based debate agents in both automatic and human evaluation, providing a stronger baseline[1]; (ii) stage-level and debate-level human evaluation is expensive (~$4,000) and time-consuming.
>
> We also tested LLaMA-3.1-8B as the backbone. However, limited capabilities in argument generation, long-context understanding, and instruction-following restrict performance in complex competitive debate. Representative failure modes for "*AI will lead to the decline of human creative arts*":
>
> | Failure Mode | Examples | Explanation |
> | --- | --- | --- |
> | Broken Logic in argument generation | "Couples reporting financial difficulties show lower satisfaction, highlighting emotional depth in art" | Logic gap between relationship satisfaction and artistic emotional depth |
> | Hallucination | "[6] ABC News (2023), Kate McCallum discussing fairness vs. equity” | Hallucinate evidence instead of using retrieved evidence |
> | Repeated Generation | [3] McCallum, K. (2023). The Future of AI Art and its Potential Interactions with Human Creativity. ABC News. … [27] McCallum, K. (2023). The Future of AI Art and its Potential Interactions with Human Creativity. … | Repeat the same reference for multiple times |
>
> These results demonstrate that 7B LLMs cannot handle competitive debate complexity. Our framework shows generalization across both open-sourced DeepSeek and closed-sourced Gemini, maintaining performance advantages with both.

---

> ### Author Response · Authors · 2025-11-21
>
> **W4: How many annotators assessed each sample? What is the inter-annotator agreement? How is bias controlled to ensure fair evaluation across different runs?**
>
> A4: Thank the reviewer for raising the important discussion on our human evaluation. We address these evaluation design choices comprehensively:
>
> - **Subjectivity Mitigation**: We do agree that debate evaluation is highly subjective, which makes the human evaluation challenging. Therefore, we design two types of human evaluation (Section 4.1)
>     - *Stage-level Head-to-Head Comparison*: All debate context remains identical, with annotators comparing two versions for a specific stage and side. This controls for prior attitudes toward the motion and previous-stage impressions.
>     - *Debate-level End-to-End Human Votes*: We use Opinion Shift Win (consistent with real Oxford-style debate), where the side gaining more vote shifts wins. Vote shift better reflects persuasiveness than raw votes, as votes are affected by prior beliefs. We flip stance assignments and average results across two competitions per motion to mitigate motion-specific bias.
> - Bias control
>     - We recruited 212 eligible participants (18+, US-based, fluent English, high school diploma+) from Prolific with diverse demographics (gender, age, political views, occupation, location; see Figure 7). We randomly sample 3 participants per comparison.
>     - Unlike deterministic problems, we do not conduct multiple runs per motion because: (i) debate has no ground-truth; outcomes depend on the audience's prior stance and preferences. It is difficult to guarantee that one debater can always beat the other on a specific motion, even for human expert debaters; (ii) overall win rates across motions are more meaningful than multiple runs of a single motion.
> - Annotators and Inter-annotator agreement
>     - **Stage-level evaluation:** We randomly recruit 3 annotators per comparison from our participant pool. Full agreement (100%) occurs when all select the same version; ties yield 50% agreement. Annotators achieved **60.7% agreement on average**, indicating moderate consensus, which is reasonable given the subjective nature of debate evaluation.
>     - **Debate-level evaluation:** We follow Oxford-style debate criteria based on vote shift (Opinion Shift Win). The winner is determined by how many annotators shift toward it, and annotators do not need to agree with each other.  Therefore, inter-annotator agreement metrics do not apply here.
>
> ---
>
> **W5: According to line 798, it appears that 52 debates are generated for the experiments. If correct, this detail should be included in the main paper. A diversity measure of debate topics**
>
> A5: We would like to clarify that 52 in Appendix C Motion List is the number of the original motions we collected. As described there, we filtered them by asking 2 human debate experts to annotate theirs polarity before our experiments to mitigate the topic bias. Table 4 lists 13 motions used in our main experiments. It shows that these motions are diverse in domains and sources.
>
> | Motion | Domain | Source |
> | --- | --- | --- |
> | Congress should abolish the debt ceiling | Economics | OpentoDebate |
> | Labor unions are beneficial to economic growth | Finance | OpentoDebate |
> | The United States should implement a central bank digital currency | Finance | OpentoDebate |
> | Processed foods should play a larger role in sustainable food systems | Health | OpentoDebate |
> | AI will lead to the decline of human creative arts | Science | OpentoDebate |
> | It is time to welcome an A.I. Tutor in the classroom | Technology | New York Times |
> | Dating Expenses Should Be Shared Equally Between Partners | Culture | New York Times |
> | Mandatory wage transparency laws should be implemented to address the gender wage gap | Economics | OpentoDebate |
> | Artists should be free to borrow from cultures other than their own | Culture | OpentoDebate |
> | If health care is a scarce resource, government should step in to ration care, deciding whose life is worth saving | Health | Oxford Dataset |
> | We should ban certain inappropriate books (like sex violence drug use) in school | Education | OpentoDebate |
> | Developed countries should impose a fat tax. | Health | Agent4debate |
> | Pursuing a four-year college degree remains beneficial for young adults in today's society | Education | New York Times |
>
> In the main experiments, we randomly selected 4 motions for Gemini and 7 for DeepSeek, conducting 2 debate games per motion (flipping stances), yielding 22 debate-level games total (Line 350).
>
> For stage-level evaluation, we randomly sampled 10 (motion, stance) settings per stage, producing 120 stage-level comparisons (Lines 342).

---

> > ### Author Response · Authors · 2025-11-21
> >
> > **Q1. In real debates, intermediate arguments are usually more complex. This tree structure might struggle to fully capture the dynamic and intricate reasoning processes that characterize authentic debates.**
> >
> > A1: We appreciate the important discussion on the complexity of the arguments in the real debate. We would like to discuss the following aspects:
> >
> > **Action complexity**: As mentioned by the reviewer, real debates involve multiple actions: attacking, defending, etc. We abstract these into four common actions: propose, attack, reinforce, and rebut (Lines 155-156, Section 3.1). Our Debate Flow Tree captures these differently:
> >
> > - **Propose:** Add a new node under the root
> > - **Reinforce:** Update existing node with new arguments
> > - **Rebut/Attack:** Add new node under target claim (demonstrating counter-argument)
> >
> > Rebut vs. attack differ in which tree they update: rebut defends against attacks in one's own tree; attack challenges the opponent's tree. These principles (Section 3.3, Algorithm 2) ensure we capture debate dynamics for planning. We extract valid candidate actions using these principles:
> >
> > - **Propose** claims at the opening stage only
> > - **Rebut** the latest claim nodes (leaf nodes) of the opponent
> > - **Reinforce** own side's claim nodes
> > - **Attack** opponent's claim nodes
> >
> > **Claim Complexity in Trees:** Figure 4 shows a simplified version. Actual nodes contain main claim, supporting arguments, evidence, status, visit count, scores, and children (Lines 221-222). Here is a realistic node example (one claim node with two children to attack this node)
> >
> > ```python
> > {
> >   "side": "against",
> >   "level": 1,
> >   "claim": "AI increases accessibility and diversity in the arts",
> >   "argument": [
> >     "AI tools enable people with limited training to create art",
> >     "A 2024 study indicates a 40% increase in art creation in underserved communities with AI",
> >     "Broadens diversity of voices, enriching cultural significance"
> >   ],
> >   "evidence": [],
> >   "status": "attacked",
> >   "visit_count": 2,
> >   "scores": null,
> >   "children": [
> >     {
> >       "side": "for",
> >       "level": 2,
> >       "claim": "AI's dominance risks losing the human connection in art by prioritizing efficiency over emotional authenticity",
> >       "argument": [
> >         "AI dominance replaces the human element, not just collaborates",
> >         "Increased art creation with AI does not enrich culture without emotional authenticity",
> >         "Opponent's case study measures clicks and views, not profound emotional connection",
> >         "Superficial interactions from AI do not equal deeper cultural impact",
> >         "Human artists draw from shared experiences that AI cannot replicate",
> >         "Prioritizing AI efficiency risks losing unpredictable, serendipitous human moments in art"
> >       ],
> >       "evidence": [],
> >       "status": "attacked",
> >       "visit_count": 3,
> >       "scores": null,
> >       "children": [
> >         {
> >           "side": "against",
> >           "level": 3,
> >           "claim": "AI art can reach a wider audience and connect with people",
> >           "argument": [
> >             "Approximately 70% of people appreciate AI art, showing positive perception",
> >             "AI art reaches a wider audience",
> >             "AI art connects with people",
> >             "AI and humans work together to create art, not replacement"
> >           ],
> >           "evidence": [],
> >           "status": "proposed",
> >           "visit_count": 1,
> >           "scores": null,
> >           "children": []
> >         },
> >         {
> >           "side": "against",
> >           "level": 3,
> >           "claim": "AI and humans work together to create art, not replacement",
> >           "argument": [
> >             "AI works together with human artists, not replaces them",
> >             "Future involves humans and machines collaborating in new ways"
> >           ],
> >           "evidence": [],
> >           "status": "proposed",
> >           "visit_count": 1,
> >           "scores": null,
> >           "children": []
> >         }
> >       ]
> >     }
> >   ]
> > },
> > ```

---

> > > ### Comment · Reviewer_hgPJ · 2025-11-28
> > >
> > > Thank you for the detailed response, which has addressed most of my earlier concerns. I also appreciate the provided sample outputs and the summary clarifying several of the points I raised. I would like to raise my overall rating to 6.0.
> > >
> > > *Seems the system does not allow us to modify the score, so I would like to inform the AC and PC of my intention to update the rating accordingly.*

---

### Official Review · Reviewer_3JDe · 2025-11-01

**Soundness:** 3
**Presentation:** 2
**Contribution:** 3
**Rating:** 4
**Confidence:** 4

**Summary:**

The paper presents TreeDebater, a framework and system for automatic debating. TreeDebater models the dynamics of a competitive debate as two trees which roughly follow the reasoning process of a human debater. The first, a Rehearsal Tree, stores a claim related to the debate topic at the root and different related arguments and counter-arguments in a tree structure below it, where each node counters its parent or enforces its grandparent. The Rehearsal Tree models the way a human debater prepares for a debate by anticipating different potential back and forth flows that may happen in the actual debate. Arguments stored in the Rehearsal Tree are scored based on the strength of their defense (of the grandparent) and attack (on the parent). The second tree, A Debate Flow Tree, follows the debate’s flow and keeps track of its status by storing in a tree structure all the claims that were made as well as related attacks and defenses that were brought up during the debate. Based on the status of the debate, the system picks the next arguments from the Rehearsal Tree. The system also includes a feedback mechanism to refine arguments and a text-to-speech system to estimate the time it takes to say a given statement, making sure that the final speech is within the allotted time. They show that humans find their system significantly more persuasive than a previous SoTA system both during certain stages of the debate as well as for the debate in its entirety.

**Strengths:**

The suggested system is inspired by the way humans prepare to and conduct a competitive debate. At a high level the system’s principles and architecture look reasonable but details are missing (see below). Evaluation includes both stage-level and end-to-end human preference experiments as well as additional fine-grained analysis of the debates that the system is producing. Experimental results are convincing.

**Weaknesses:**

Many details are missing. For example:

- It is not clear what is the action selection criteria in the paragraph starting in line 227: “Extract Candidate Actions from Debate Flow Tree”.

- When updating the Debate Flow Tree (alg. 2 in Appendix B), how is the ‘action’ being determined?

- Even after reading Appendix D, it is not clear to me how the audience feedback works.

**Questions:**

- Line 418: how do you determine if two claims are similar?

- What is the agreement between the annotators in the human preference experiments?

- Line 368: Figure 2 -> Table 2

---

> ### Author Response · Authors · 2025-11-21
>
> We appreciate the reviewer’s knowledge of our human-inspired tree structure in competitive debate. The constructive comments help us strengthen the manuscript. We have revised our manuscript accordingly (highlighted in green) and addressed the remaining concerns below.
>
> **W1: It is not clear what are the action selection criteria in the paragraph starting in line 227: “Extract Candidate Actions from Debate Flow Tree”.**
>
> A1: At each debate stage, suppose the debater raises M claims and the opponent raises N claims. With 4 atomic actions per claim (propose, attack, reinforce, rebut), there are 4(M+N) possible (action, claim) combinations. We extract valid candidate actions using these principles:
>
> - **Propose** claims at the opening stage only
> - **Rebut** the latest claim nodes (leaf nodes) of the opponent
> - **Reinforce** one's own side's claim nodes
> - **Attack** opponent's claim nodes
>
> These principles are now fully detailed in Section 3.3 Debate Flow Tree.
>
> ---
>
> **W2: When updating the Debate Flow Tree (Alg. 2 in Appendix B), how is the ‘action’ being determined?**
>
> A2: To get (action, claim, argu, target) in Alg. 2, we use the backbone LLM to extract the tuples from the stage statement. The simplified prompt is shown below:
>
> ```markdown
> ## Task: Analyze the statements
> Your task is to analyze the statements and identify the key claims presented in the statement and the evidence or reasoning to support the claims.
> You are given two debate trees that model the back-and-forth between you and your opponent. Your extracted claims can be used to:
> 	- propose the main claims under Level-0 of your debate tree
> 	- rebut the opponent's attacks in Level-2 of your debate tree.
> 	- reinforce the main claims in Level-1 of your debate tree.
> 	- attack the opponent's proposed claims in Level-1 of your opponent's debate tree.
> Each claim should be used for one of the above purposes or a combination of them.
>
> ## Input Information
> **Debate Topic**: {motion}
> **Your Stance**: {side}
> **Current Stage**: {stage}
> **Statement**: {statement}
> **Your Debate Tree**:
> {tree}
> **Opponent's Debate Tree**:
> {oppo_tree}
>
> ## Response Format
> Provide your response in JSON format with one key of **statements**.  The keys of each element of the list are **claim**, **content**, **type**, **arguments**, **purpose**.
> ```
>
> The full prompt can also be found in our anonymous code: https://anonymous.4open.science/r/debate-anonymous-40F1/src/utils/prompts/others.py.
>
> ---
>
> **W3: Even after reading Appendix D, it is not clear to me how the audience feedback works.**
>
> A2: Thanks for raising this important point. We clarify the human debate corpus construction and retrieval process as below, and have revised the manuscript to make it clear (Lines 234-245,
>
> - **Build Human Debate Flow Tree Corpus**: We extract debate transcripts from PanelBench and reorganize them into the Oxford debate format (opening, rebuttal, closing). For each phase, we extract (action, claim, argument, target claim) tuples via LLM and construct the Debate Flow Tree using Algorithm 2 (Page 15).
> - **Retrieve from Human Debate Flow Tree Corpus**: We convert the current Debate Flow Tree into a tree-like string and perform semantic search using Gemini-text-embedding-4. We retrieve the top human debate tree (also in a tree-like string) with a similarity threshold of 0.8.
> - **Audience Feedback with Human Debate Flow Tree:** The simulated audience receives the retrieved human debate example to understand how similar debate topics were handled. Here is a simplified example:
>
> ```python
> """
> Level-0 Motion: AI will lead to the decline of human creative arts, Side: for
>     Level-1 Your Main Claim: AI-generated content lacks the emotional depth and soul of human-created art (arguments: [...])
>         Level-2 Opponent's Attack: AI enhances human creativity by providing new tools and inspiration (arguments: [...])
>     Level-1 Your Main Claim: AI causes economic devaluation of human art by flooding the market and driving down prices (arguments: [...])
>         Level-2 Opponent's Attack: AI supports human artists economically by creating new revenue streams (arguments: [...])
>     Level-1 Your Main Claim: AI lacks the ability to produce true artistic breakthroughs and originality due to reliance on existing data (arguments: [...])
>         Level-2 Opponent's Attack: AI fosters new forms of artistic expression and originality (arguments: [...])
>     Level-1 Your Main Claim: Decline in human creative arts refers to a reduction in quality, economic value, and cultural significance (arguments: [...])
>     Level-1 Your Main Claim: AI's dominance in creative spaces risks losing the essential human connection in art (arguments: [...])
>         Level-2 Opponent's Attack: AI strengthens human connection in art through collaborative projects (arguments: [...])
> """
> ```
>
> Table 15 provides a concrete audience feedback example.

---

> > ### Author Response · Authors · 2025-11-21
> >
> > **Q1: Line 418: How do you determine if two claims are similar?**
> >
> > A1: We use Gemini-text-embedding-4 to obtain claim embeddings and calculate cosine similarity between them. Claims are considered similar if similarity exceeds a threshold of 0.8. This threshold balances relevance and coverage, established through pilot studies. This detail was described in Section B.1 and has been moved to the main body (Lines 241-245).
> >
> > ---
> >
> > **Q2: What is the agreement between the annotators in the human preference experiments?**
> >
> > A2: **Stage-level evaluation:** We randomly recruit 3 annotators per comparison from our online participant pools. Full agreement (100%) occurs when all select the same version; ties yield 50% agreement. Annotators achieved **60.7% agreement on average**, indicating moderate consensus, which is reasonable given the subjective nature of debate evaluation.
> >
> > **Debate-level evaluation:** We follow Oxford-style debate criteria based on vote shift (Opinion Shift Win). The winner is determined by how many annotators shift toward it, and annotators do not need to agree with each other.  Therefore, inter-annotator agreement metrics do not apply here.

---

> > ### Comment · Reviewer_3JDe · 2025-11-25
> >
> > Thank you for the clarifications. I raised the rating to 6.

---

> > > ### Author Response · Authors · 2025-11-26
> > >
> > > Thank you very much for raising the score! Any further discussion is more than welcome.

---

### Official Review · Reviewer_eGTP · 2025-11-01

**Soundness:** 3
**Presentation:** 2
**Contribution:** 3
**Rating:** 4
**Confidence:** 3

**Summary:**

This work tackles the problem of automated competitive debate, and focuses on the aspect of planning and decision making. The authors enhance an existing agent system for debating by adding capabilities of anticipating the expected flow and claims that will be made by both sides, and tracking the flow as the debate progresses. This is done via constructing trees where each node is an argument or counter-argument. Paths are chosen based on their estimated utility, in terms of the chosen action (e.g., reinforce an existing claim or rebut an opponent claim), the estimated argument strength, and semantic similarity is used to retrieve prepared arguments and track how many times an argument is addressed. The authors conduct experiments comparing their approach to the baseline agent debate system that does not incorporate this tree-based planning, performing human evaluation of per-stage debate persuasiveness and performance as well as a full head-to-head debate between the two systems.

**Strengths:**

1. The tree-based methods are novel and interesting. In particular, it has a nice approach of estimating the argument strength not just in terms of the argument itself but in terms of anticipating its full impact including the estimated opponent actions that will follow.
2. Human evaluation was done at the level of a specific stage as well as the overall debate impact, and shows some persuasiveness gains from the proposed approach.

**Weaknesses:**

1. There are many details here and no provided code, so reproducibility is a major issue. Moreover, given that the paper is focused on a comparison to Agent4Debate, I think it is missing a clearer accounting of how and in what architecture the method here integrates with the agent system described there. The diagram in Figure 1 is helpful but is quite vague in terms of understanding how the LLM agents are used in practice when generating the debate.
2. I felt that the part about simulated audience feedback (§3.4) was not sufficiently clear. Without going to the appendix, it is not explicitly stated that there is a collected dataset of human debates (the reference to "the retrieved human Debate Flow Trees" (l. 249) is too vague on its own). Crucially, I did not understand from either the main paper or the appendix how exactly this data is used.

**Questions:**

1. Could you explain where the list of "battlefields" included in the prompts comes from?
2. In Figure 2, what is the difference in implementation between the baseline Agent4Debate (which AFAIU doesn't use trees) and the ablation of TreeDebater without the trees? Why do they show such different behaviors?
3. To what extent would you say that the ability to anticipate the opponent behavior (Figure 3) is connected to the similarity between the two automated systems engaged in the debate (e.g., that both use the same underlying LLM and similar prompts to generate the arguments)?
4. What is the value of the decay coefficient $\gamma$?

Additional comments:
* I think it would help to highlight in the prompts (Appendix G) which parts of the prompt comes from Agent4Debate versus are new for this work.
* Appendix D mentions "two debate datasets", but then only PanelBench is mentioned.

Typos:

l. 33 argued -> argue

l. 167 this grandparent -> its grandparent

l. 240/295/302/357 the Appendix -> Appendix

l. 348 totally recruited 212 participants -> recruited 212 participants in total

---

> ### Author Response · Authors · 2025-11-21
>
> We sincerely thank the reviewer for the thorough and constructive feedback. We are deeply grateful for recognizing the novelty of our tree-based debate planning, particularly the incorporation of opponent behavior simulation in argument strength estimation. We have revised our manuscript accordingly (highlighted in green) and address the remaining concerns below.
>
> **W1: Missing details and reproducibility. Missing a clearer accounting of how and in what architecture the method here integrates with the agent system described there.**
>
> A1: Thanks for the suggestion. We provide an anonymous link to upload our code (https://anonymous.4open.science/r/debate-anonymous-40F1) and will release it after acceptance. The code and detailed README address reproducibility concerns.
>
> We selectively reuse debate-related prompts from Agent4Debate:
>
> - Stage-specific prompts from Agent4Debate's **writer** to our **writer** module (Figure 1)
> - The **searcher** prompt to our **Rehearsal Tree for evidence search** with Tavily APIs (Lines 293-295)
>
> TreeDebater significantly differs from Agent4Debate:
>
> |  | Agent4Debater | TreeDebater |
> | --- | --- | --- |
> | Architecture | writer, searcher, analyzer, reviewer | writer, simulated audience |
> | Planning | implicit planning (analyzer) | explicit planning (Rehearsal Tree, Debate Flow Tree) |
> | Memory | none | human debate corpus |
> | Reflection | feedback (reviewer) | simulated audience feedback + time controller |
> | Time control | none | speech time controller |
>
> The statement generation pipeline (Figure 1, detailed in Section 3) uses LLMs as follows:
>
> | Steps | LLM usage | Tools |
> | --- | --- | --- |
> | Update Debate Flow Tree | extract (action, target, claim, argument) tuples from the opponent’s statement |  |
> | Retrieve prepared arguments from Rehearsal Trees | / | semantic similarity retrieval |
> | Generate statement from prepared arguments | Generate statement |  |
> | Retrieve similar human debate flow tree | / | semantic similarity retrieval |
> | Audience feedback | Generate feedback using retrieved example |  |
> | Speech time controller | / | FastSpeech time estimation |
> | Iterative revision | Revise statement based on feedback and time |  |
>
> ---
>
> **W2: Simulated audience feedback (§3.4) was not sufficiently clear.**
>
> A2: We clarify the human debate corpus construction and retrieval process:
>
> - **Build Human Debate Flow Tree Corpus**: We extract debate transcripts from PanelBench and reorganize them into the Oxford debate format (opening, rebuttal, closing). For each phase, we extract (action, claim, argument, target claim) tuples via LLM and construct the Debate Flow Tree using Algorithm 2 (Page 15).
> - **Retrieve from Human Debate Flow Tree Corpus**: We convert the current Debate Flow Tree into a tree-like string and perform semantic search using Gemini-text-embedding-4. We retrieve the top human debate tree (also in a tree-like string) with a similarity threshold of 0.8.
> - **Audience Feedback with Human Debate Flow Tree:** The simulated audience receives the retrieved human debate example to understand how similar debate topics were handled. Here is a simplified example:
>
> ```python
> """
> Level-0 Motion: AI will lead to the decline of human creative arts, Side: for
>     Level-1 Your Main Claim: AI-generated content lacks the emotional depth and soul of human-created art (arguments: [...])
>         Level-2 Opponent's Attack: AI enhances human creativity by providing new tools and inspiration (arguments: [...])
>     Level-1 Your Main Claim: AI causes economic devaluation of human art by flooding the market and driving down prices (arguments: [...])
>         Level-2 Opponent's Attack: AI supports human artists economically by creating new revenue streams (arguments: [...])
>     Level-1 Your Main Claim: AI lacks the ability to produce true artistic breakthroughs and originality due to reliance on existing data (arguments: [...])
>         Level-2 Opponent's Attack: AI fosters new forms of artistic expression and originality (arguments: [...])
>     Level-1 Your Main Claim: Decline in human creative arts refers to reduction in quality, economic value, and cultural significance (arguments: [...])
>     Level-1 Your Main Claim: AI's dominance in creative spaces risks losing the essential human connection in art (arguments: [...])
>         Level-2 Opponent's Attack: AI strengthens human connection in art through collaborative projects (arguments: [...])
> """
> ```
>
> Table 15 provides a concrete audience feedback example.
>
> ---

---

> > ### Author Response · Authors · 2025-11-21
> >
> > **Q1: "battlefields" included in the prompts comes from?**
> >
> > A1: Thank you for raising this point. We defined the battlefield in Debate Terminology in Appendix A, which refers to a conflict zone in the debate, e.g. two sides contest on a specific point. It includes several rounds of propose, attack, and defense. In the tree structure, it is usually a subtree with several claim nodes. In practice, we view the tuples of (action, claim, argument, target claim) as one battlefield. We have moved this definition to our main body (Lines 156-157).
> >
> > ---
> >
> > **Q2: In Figure 2, what is the difference in implementation between the baseline Agent4Debate (which AFAIU doesn't use trees) and the ablation of TreeDebater without the trees? Why do they show such different behaviors?**
> >
> > A2. Please refer to the comparison table in W1. Without the tree structures, ablated TreeDebater retains only the writer and simulated audience modules. Agent4Debate maintains implicit planning through its analyzer agent, while TreeDebater w/o trees lacks any planning module. This absence of planning produces fewer diverse debate actions compared to Agent4Debate.
> >
> > ---
> >
> > **Q3: To what extent would you say that the ability to anticipate the opponent's behavior (Figure 3) is connected to the similarity between the two automated systems engaged in the debate (e.g., that both use the same underlying LLM and similar prompts to generate the arguments)?**
> >
> > A3: We analyze how often TreeDebater correctly anticipates opponent behavior. We define hit rate as # found opponent claims in Rehearsal Tree / # opponent claims, varying both backbone LLMs and agent systems:
> >
> > | Debater A | Debater B | Hit Rate of Debater A’s  |
> > | --- | --- | --- |
> > | TreeDebater (DeepSeek) | TreeDebater (DeepSeek) | 69.8% |
> > | TreeDebater (DeepSeek) | Agent4Debate (DeepSeek) | 72.0% |
> > | TreeDebater (DeepSeek) | TreeDebater (Gemini) | 61.7% |
> > | TreeDebater (DeepSeek) | Agent4Debate (Gemini) | 67.7% |
> >
> > Hit rates are higher with the same backbone LLM. Interestingly, TreeDebater anticipates Agent4Debate better than itself. This suggests that Agent4Debate's simpler, prompt-based planning behavior (without explicit planning) is easier to predict during the creation of the rehearsal tree. However, TreeDebater's complex Debate Flow Tree planning produces harder-to-predict behavior.
> >
> > ---
> >
> > **Q4: What is the value of the decay coefficient?**
> >
> > A4: We use a decay coefficient of 0.8, now included in our revision (Line 188).
> >
> > ---
> >
> > **Q5: Additional comments on the prompt highlights. Appendix D mentions "two debate datasets", but then only PanelBench is mentioned.**
> >
> > A5: Thanks for the valuable suggestions. We corrected these issues and highlighted prompts borrowed from Agent4Debate (Tables 12, 13, 14). The "two debate datasets" refer to one dataset (PanelBench) collected from two sources: DebateArt and BP-Competition. We have clarified this in the revision.

---

> ### Comment · Reviewer_eGTP · 2025-11-24
>
> Thank you for your detailed response to my comments!
>
> I greatly appreciate your willingness to release the code. In my mind, this goes a long way to making the work more reproducible and impactful.
>
> At the same time, to clarify things for the readers my feeling is that the method descriptions in the text still could use a few clarifications:
> * Your response was very helpful in explaining the overall architecture, and the differentiation vs. Agent4Debate - IIUC, here you have a relatively simple setup (which is a good thing!) where the main generation nodes are just a writer node that generates the statements and an additional feedback node, along with a more elaborate tree-based planning setup that helps the writer node do its job. In my view it would help avoid confusion to state explicitly in the paper that this is the overall architecture (e.g., in the caption of Figure 1).
> * While I see you have added more information on the retrieval of a human debate tree for the simulated audience feedback, the paper still does not explicitly explain what this is used for. I assume there is some reflection prompt that says something like "please provide feedback on the following aspects… here is an example of a good human debate for reference {retrieved_human_debate}". Is my understanding correct? Providing and linking to the relevant prompt could help clarify to the reader what is being done here. To me, the name given of "simulated audience" implies that there is something more sophisticated to this component than just asking an LLM to give feedback, but I do not understand where is the "audience simulation" aspect versus just a standard LLM reflection node.
>
> The analysis you gave in response my Q3 is very interesting in my opinion.

---

> > ### Author Response · Authors · 2025-11-24
> >
> > Thank the reviewer for the active discussion! We are glad to hear that our released code is helpful in reproduction, and our explanations on the difference with Agent4Debate make our method clearer and easier to understand. As suggested, we have revised the description in the caption of Figure 1 to avoid confusion.
> >
> > For the simulated audience, yes, we use a similar instruction to guide the audience feedback with the retrieved human debate flow tree, which is shown below. Besides, this simulated audience is different from the LLM reflection:
> >
> > - **Have no access to the debater’s internal thoughts.** The individual simulated audience agent provides feedback only based on what it hears, without knowing the debater’s internal planning (such as the Debate Flow Tree, the retrieval arguments with strength scores, etc.). This aligns with the audience behavior in human competitive debate, which only has access to the final statements from both sides.  Instead, the LLM reflection has access to all internal reasoning and provides feedback based on all of these.
> >     - e.g., In *Evidence Presentation,* the simulated audience judges solely based on what evidence is presented in the final statement. However, the LLM reflection can see all internal reasonings, including all retrieved arguments and evidence from the Rehearsal Tree. This makes LLM reflection less aware of the lack of evidence in the final statement.
> > - **Have access to external human debate flow tree**. The LLM reflection relies on its own judgment, while the simulated audience provides feedback based on human debate.
> > - **Have different audience profiles.** As shown below, we define a diverse audience panel composition to help the simulated audience provide more diverse feedback, considering different audiences.
> >
> > The full prompt can be found in Lines 277-352 of https://anonymous.4open.science/r/debate-anonymous-40F1/src/utils/prompts/others.py. And we summarize the core part below:
> >
> > ```json
> > ## Your Task
> > You are a panel of debate audience members to provide comprehensive feedback on how the statement impacts and persuades a general audience.
> >
> > ### Audience Panel Composition
> > - General public with varied educational backgrounds
> > - Students and educators from different fields
> > - Professionals interested in policy and social issues
> >
> > ### Evaluation Dimensions
> > 1. **Core Message Clarity**
> > 2. **Engagement Impact**
> > 3. **Evidence Presentation**
> > 4. **Persuasive Elements**
> >
> > ### Guidelines
> > - Evaluate all dimensions thoroughly
> > - Identify the most significant barriers to audience understanding in the {stage} statement
> > - Consider which issues could be addressed with minimal revisions on the {stage} statement
> > - Focus on high-impact, low-disruption improvements
> >
> > ## Retrieval Information
> > Here are debate flow trees and action allocations from human debates.
> > Use the structure and allocation strategy to provide better feedback.
> >
> > {retrieval_debate_flow_tree}
> >
> > ### Input Information
> > **Debate Topic**:
> > {motion}
> > **History of the debate**:
> > {history}
> > **Current {side}'s {stage} Statement to be evaluated**:
> > {statement}
> >
> > ### Output Format
> > [Comprehensive Analysis]
> > Core Message Clarity:
> > Engagement Impact:
> > Evidence Presentation:
> > Persuasive Elements:
> > [Critical Issues and Minimal Revision Suggestions]
> > Issue:
> > Impact on Audience:
> > Minimal Revision Suggestion:
> > ```
> >
> >
> > Thank you for your thoughtful feedback. We welcome any additional comments or discussion.

---

### Official Review · Reviewer_FvEB · 2025-11-01

**Soundness:** 3
**Presentation:** 3
**Contribution:** 3
**Rating:** 6
**Confidence:** 3

**Summary:**

The paper proposes TreeDebater, an LLM-based debating system designed for competitive, time‑limited debates. The core idea is to structure planning and tracking as two trees:

Rehearsal Tree (pre‑debate): based on the intuition that human debaters usually prepare for the potential attacks and defenses of
their claims. The proposed rehearsal tree is designed to help the debator retrieve relevant evidence and evaluate how robust their claims are towards the attack. The Rehearsal Tree anticipates the attack and defense for each main claim in a tree format and calculates the k-step strength score to evaluate the utility of the claim.

Debate Flow Tree (in‑debate): The debate tree is designed to record the debate status, simulating the note-taking of humans. The Debate Flow Tree tracks the debate status by keeping all proposed claims with the corresponding attack and defense in a tree structure. TreeDebater can filter out the candidate actions it can take in the current stage of the debate based on the Debate Flow Tree. After getting the candidate actions, TreeDebater retrieves the prepared arguments for this action from the Rehearsal Trees.

Empirically, this paper compares TreeDebater with Agent4Debate across two backbones (Gemini‑2.0‑flash and DeepSeek‑V3) using human evaluation at (a) the stage level (head‑to‑head on the same debate context) and (b) the end‑to‑end debate level with pre/post votes. TreeDebater improves average persuasiveness and opinion‑shift win rate.

**Strengths:**

1. The paper tackles competitive debate where persuasiveness determines success, and it operationalizes time budgeting and action selection under evolving interaction. The two‑tree design mirrors how human debaters prepare and flow debates.
2. The Rehearsal Tree formalizes pre‑debate planning with a k‑step strength score that blends support and attack impacts.
3. The Debate Flow Tree offers a practical, auditable representation for candidate action extraction during the match.
4. Good empirical evaluation to demonstrate the effectiveness of the proposed approach.
5. The paper is well written and easy to follow.

**Weaknesses:**

1. TreeDebater alone uses an iterative time controller; the baseline is given only “rough word budgets” and is then audio‑trimmed, which can truncate arguments mid‑point and plausibly depress persuasiveness.
2. Missing detailed ablations for each component of the TreeDebater in achieving the final persuasiveness. For example, the impact of Rehearsal Tree and Debate Flow Tree separately is unknown.
3. While the overall pipeline is well designed, the fundamental idea shares similarity with Tree‑of‑Thoughts / Graph‑of‑Thoughts.

**Questions:**

N/A

---

> ### Author Response · Authors · 2025-11-21
>
> We appreciate the reviewer’s positive opinions on the effectiveness of the Rehearsal Tree and Debate Flow Tree in our debate agent. Further insightful comments help us improve the quality of our manuscript. We have revised our manuscript accordingly (highlighted in green) and addressed the remaining concerns below.
>
> **W1: The baseline is given only “rough word budgets” and is then audio‑trimmed, which can truncate arguments mid‑point and plausibly depress persuasiveness.**
>
> A1: We agree that audio truncation may affect persuasiveness—this actually reflects real competitive debate where speakers must decide which points to keep under time constraints. That is one of the main limitations of the Agent4Debate baseline when applied to the real-world competitive debate.
>
> To address this, we introduce the **speech time controller** (Lines 081-082) to enable the agent to appropriately assign time budgets while maintaining persuasiveness. For the Agent4Debate baseline, we strengthen it by adding explicit word budget instructions to make it time-aware for a fair comparison
>
> ---
>
> **W2: Missing detailed ablations for each component of the TreeDebater in achieving the final persuasiveness.**
>
> A2: We further conduct the head-to-head evaluation on the ablation of the Rehearsal Tree and Debate Flow Tree. We select two new motions and use the Gemini model as the backbone. The results are shown below.
>
> | Framework | Opening | Rebuttal | Closing |
> | --- | --- | --- | --- |
> | TreeDebater | 3.50 | 3.50 | 3.75 |
> | TreeDebater w/o Rehearsal Tree | 3.00 | 3.25 | 3.50 |
> | TreeDebater w/o Rehearsal & Debater Flow Tree | 3.00 | 3.00 | 3.50 |
>
> Results show clear performance degradation when removing components, confirming their necessity for better debate quality.
>
> The simulated audience provides feedback on **message clarity, engagement, evidence quality, and persuasiveness** by comparing statements against real human debate (Section 3.4). This feedback helps TreeDebater refine the presentation. Qualitative analysis (Table 4) shows refinements improve audience awareness and emotional tone, making arguments more convincing. See Table 15 (Page 26) for an example.
>
> ---
>
> **W3: While the overall pipeline is well designed, the fundamental idea shares similarity with Tree‑of‑Thoughts / Graph‑of‑Thoughts.**
>
> A3: While both Tree‑of‑Thoughts (ToT) / Graph‑of‑Thoughts (GoT) also use tree structures, they cannot be applied to competitive debate because:
>
> - ToT and GoT only model LLMs’ internal reasoning, while in competitive debate, the interaction between two sides requires the agent models both its own and the opponent’s behaviors
> - ToT/GoT node’s score only depends on its own correctness, while the claim node's score in TreeDebate depends on its own claim quality and the potential attack from the opponent. Therefore, TreeDebater uses a minimax-like scoring to calculate the node’s strength by anticipating the opponent’s behaviors in the Rehearsal Tree, and keeps a dynamic attack-defense relation in the Debate Flow Tree to track actual behaviors from both sides.
>
> To sum, ToT/GoT and TreeDebater differ fundamentally because:
>
> | **Aspect** | ToT / GoT | TreeDebater |
> | --- | --- | --- |
> | **Node**  | Intermediate reasoning steps | Claims/arguments from both sides |
> | **Edge** | Chronological reasoning order | How claims counter each other |
> | **Construction** | Propose alternative next steps as branches | Rehearsal Tree: simulate opponent behaviors (Algorithm 1). Debate Flow Tree: extract both sides' actions (Algorithm 2) |
> | **Usage** | Explore reasoning paths for problem-solving | Rehearsal Tree: anticipate claim strength through simulated interaction. Debate Flow Tree: track back-and-forth debate dynamics |
>
> **Key difference:** ToT/GoT structures contain only the agent's reasoning for single-agent exploration. TreeDebater considers both agents' claims to model interactive back-and-forth, which is naturally more complex

---

### Author Response · Authors · 2025-12-03
**Thanks ACs for their hard work and here is a summary comment for the discussion phase**

We sincerely thank the reviewers for the thorough and constructive feedback, which helps us improve the quality of our manuscript.

We are grateful to see that the reviewers think

- Our tree-based debate planning is novel and interesting [Reviewer **eGTP**]
- Our Rehearsal Tree and Debate Flow Tree is well-motivated by human debating [Reviewer **hgPJ,3JDe**]
- Our paper is well-written and easy to follow [Reviewer **FvEB**]
- Experimental results are comprehensive and convincing [Reviewer **FvEB, hgPJ, 3JDe**]

---

We really appreciate reviewers’ engagement in discussion, and are happy to see that

- Reviewer **3JDe** and **hgPJ** all thought our responses had addressed their concerns and both decided to **raise their score to 6.**
- Reviewer **eGTP** thought the released code made our work more reproducible and impactful, and found our response very helpful. As the reviewer further requested, we have added more demonstrations to show our prompt for the simulated audience feedback.

---

During the discussion, we have systematically addressed the concerns mentioned by reviewers, including:

- Concern 1: Comparison between our TreeDebater and reasoning works such as ToT / GoT [Reviewer **FvEB, hgPJ**]
    - Response: We focus on the unique challenges brought by competitive debate: **strategic choices about which points to state** in back-and-forth interaction **between one and their opponent** with a **limited time**
    - Response: Our Tree Structure incorporates the anticipated **opponent’s behaviors** to estimate the minimax score of each node, which differs fundamentally with ToT
- Concern 2:  Reproduction and implementation details [Reviewer **eGTP, 3JDe, hgPJ**]
    - Response: We uploaded our anonymous code: https://anonymous.4open.science/r/debate-anonymous-40F1
    - Response: We provided the important details about the simulated audience from the appendix to **Section 3.4** in the main body
    - We added more examples and prompt templates for better demonstration
- Concern 3: More detailed ablation studies [Reviewer **FvEB, hgPJ**]
    - Response: We conducted **additional ablation human evaluation** on our key components, demonstrating their effectiveness
- Concern4: Inter-agreement of Human Evaluation [Reviewer **3JDe, hgPJ**]
    - Response: Annotators achieved **60.7% agreement on average**, indicating moderate consensus, which is reasonable given the subjective nature of debate evaluation.

---

We have revised our manuscript based on these valuable suggestions (the revised part is highlighted in green), including

- Clarify the definition of *battlefield* in **Section 3.1**
- Add details of action extraction and selection in **Section 3.3**
- Move details of simulated audience from Appendix to **Section 3.4**
- Add the inter-annotator agreement to **Section 4.1**
- Add a fine-grained ablation study to **Section 4.3**
- Add more details in the human debate flow tree corpus in **Appendix D**
- Modify the prompt templates in **Table 12,13,14** for better demonstration
- Fix some typos

---

We sincerely appreciate the reviewers' collaborative engagement and hope our contributions and responses receive full consideration.

---

### Meta-Review · Area_Chair_dEKw · 2025-12-10

**Summary:**

While this paper originally faced substantial drawbacks, it seems that the rebuttal included substantial work, especially in terms of clarifying, releasing code and replicability issues. As the changes are so major I do encourage the paper to have a clean read of all the changes and also ensure that any question answered here is updated in the corresponding place in the paper.

**Reviewer Concerns:**

The reviewers had various concerns, many of which were about clarity. It seems also most of them were addressed.

**Reviewer Scores:**

That is not a fair, relevant or meaningful question. I protest the way this was all handled.
A Reviewers are not here, and ToM is weak, at least mine and the one literature study. I will not try to predict people.
B Scores are anyway a weak signal of interest, a paper should not accepted or rejected just based on it, that's an ACs job to look at the specific weaknesses and translate it into a recommendation.
C There are about 100 pages of discussions for me to read overall, in addition to the discussions I monitored and were just replaced, this is beyond my personal ability to do fairly. I did my best effort.

Still, it seems that in this case, reviewers were quite likely to change their minds or at the very least to agree that most issues in the paper are minor.

---

### Decision · Program_Chairs · 2026-01-26

Accept (Poster)